# A simple model of the turnover of organic carbon in a soil profile: model test, parameter identification and sensitivity

Elsa Coucheney[1], Anke Marianne Herrmann[1], Nicholas Jarvis[1]

[1]Department of Soil and Environment, Swedish University of Agricultural Sciences, Lennart Hjelms väg 9, 756 51 Uppsala, Sweden

*Correspondence to*: Elsa Coucheney (elsa.coucheney@slu.se)

**Abstract.** Simulation models are potentially useful tools to test our understanding of the processes involved in the turnover of soil organic carbon (SOC) and to evaluate the role of management practices in maintaining stocks of SOC. We describe here a simple model of SOC turnover at the soil profile scale that accounts for two key processes determining SOC persistence (i.e. microbial energy limitation and physical protection due to soil aggregation). We tested the model and evaluated the identifiability of key parameters using topsoil SOC contents measured in three treatments with contrasting organic matter inputs (i.e. fallow, mineral fertilized and cropped, with and without straw addition) in a long-term field trial. The estimated total input of organic matter (OM) in the treatment with straw added was roughly three times that of the treatment without straw addition, but only 12% of the additional OM input remained in the soil after 54 years. By taking microbial energy limitation and enhanced physical protection of root residues into account, the model could explain the differences in C persistence among the three treatments, whilst also accurately matching the time-courses of SOC contents using the same set of model parameters. Models that do not explicitly consider microbial energy limitation and physical protection would need to adjust their parameter values (either decomposition rate constants or the retention coefficient) to match this data.

We also performed a sensitivity analysis to identify the most influential parameters in the model determining soil profile stocks of OM at steady-state. Input distributions for soil and crop parameters in the model were defined for the agricultural production region in east-central Sweden that includes Uppsala. This analysis showed that model parameters affecting SOC decomposition rates, including the rate constant for microbial-processed SOC and the parameters regulating physical protection and microbial energy limitation, are more sensitive than parameters determining OM inputs. The development of pedotransfer approaches to estimate SOC decomposition rates from soil properties would therefore support predictive applications of the model at larger spatial scales.

**Copyright statement.**

## 1    Introduction

Adopting soil and crop management practices that increase stocks of soil organic carbon (SOC) is one promising way to mitigate climate change, whilst simultaneously improving soil health (Paustian et al., 2016; Baveye et al., 2020). In conjunction with long-term field experiments, simulation models are useful tools for testing our understanding of the processes involved in the turnover of SOC and for evaluating the potential of management practices to enhance SOC sequestration. Most model applications to date have focused on cultivated topsoil, which

is clearly of major importance with respect to the effects of soil management on SOC and soil health. However, subsoils contain a large proportion of the total stock of SOC (Batjes, 1996; Jobbágy and Jackson, 2000; Poeplau et al., 2020) and residence times are also much longer (Rumpel and Kögel-Knabner, 2011; Sierra et al., 2024; Button et al., 2022). This may indicate a significant potential for long-term C sequestration of root-derived OM in subsoils, which could be of substantial benefit in mitigating climate change.

Several detailed mechanistic models have been developed that describe a wide range of processes affecting C stocks at the scale of the entire soil profile, including soil water flow, transport of dissolved organic carbon by advection-diffusion and bioturbation, as well as descriptions of SOC decomposition explicitly accounting for microbial processes (e.g. Izaurralde et al., 2006; Braakhekke et al., 2011; Riley et al., 2014; Ahrens et al., 2015; Camino-Serrano et al., 2018; Hicks Pries et al., 2018; Keyvanshokouhi et al., 2019; Yu et al., 2020). Such

mechanistic models are useful tools for improving process understanding (Smith et al., 2018; Derrien et al., 2023), but parameter uncertainty and the ever-present likelihood of equifinality means that predictive model applications may be problematic (Braakhekke et al., 2013). Simpler empirical (phenomenological) models of SOC turnover and storage may have an advantage in this respect because they require fewer parameters (Derrien et al, 2023).

Although simple models are in principle well suited to policy and management applications, their validation status

is generally poor: many have been extensively calibrated against field observations, but their reliability in extrapolation (i.e. prediction of independent data) has not yet been convincingly demonstrated (Garsia et al., 2023; Le Noë et al., 2023). This is because these models have often been tested against limited datasets (i.e. observations of topsoil C dynamics at a single site and treatment) which increases the likelihood of equifinality despite the small number of parameters (e.g. Juston et al., 2010; Luo et al., 2017). This may be overcome by simultaneous

calibration of the model against data for two or more contrasting treatments, for example with respect to the type and quantity of organic matter inputs (e.g. Meurer et al., 2020) or by multi-site calibration at larger scales using data from long-term field trials at locations with contrasting soils and management practices (e.g. Juston et al., 2010; Dechow et al., 2019). Testing model predictions for entire soil profiles remains however difficult and is therefore rarely done, because fewer measurements are made in subsoils and the turnover of organic C in subsoil

is very slow, so datasets will rarely be long enough to detect any changes (Balesdent et al., 2018). Additional data sources may also help to alleviate problems arising from equifinality. One possibility is to make use of $^{14}$C concentrations as a measure of SOC age (e.g. Braakhekke et al., 2014; Ahrens et al., 2015; Sierra et al., 2018) or concentrations of natural stable isotopes of C (Balesdent and Mariotti, 1987), their ratio $^{12}$C/$^{13}$C in C3-C4 vegetation chronosequences (Schiedung et al., 2017; Balesdent et al., 2018) or labelled material (Sanaullah et al.,

2011). If such data is missing, an alternative approach to model validation is to compare model predictions against spatial (soil survey) datasets either at catchment, regional or national scales. This has often been done for the topsoil (e.g. Sleutel, et al., 2006; Yagasaki and Shirato, 2014), but to our knowledge there are no examples of this approach in the published literature dealing with total stocks of organic C in the profile.

Ideally, a model that is intended for predictive management applications at the soil profile scale should combine the advantages of simplicity with descriptions that adequately capture or mimic the most important processes determining SOC stocks (Campbell and Paustian, 2015). In this respect, using a more complex process-oriented model, Sierra et al. (2024) recently concluded that DOC transport and bioturbation are generally only of limited importance for subsoil SOC stocks, which are instead largely determined by the balance between root-derived inputs and decomposition rates. In turn, experimental evidence suggests that decomposition rates of SOC are affected mostly by bioavailability (i.e. soil properties controlling adsorption; Mathieu et al., 2015), physical protection (e.g. Killham et al., 1993; Strong et al., 2004; Salomé et al., 2010) and the amount of SOC as it provides energy for microbial biomass growth, maintenance and activity (e.g. Fontaine et al., 2007; Don et al., 2013; Wutzler and Reichstein, 2013). We are not aware of any relatively parsimonious (or minimalist) model that has been shown to capture the effects of these key processes on SOC stocks at the scale of an entire soil profile.

The overall aim of this study is to demonstrate the utility of a simple soil C turnover model that that is specifically designed to fill this gap by accounting for the nexus of soil management, soil structure and microbial activity that critically determines C mineralization and stabilization at the scale of a soil profile. The model structure is based on ICBM (**I**ntroductory **C**arbon **B**alance **M**odel; Andrén and Kätterer, 1997), which contains two C pools (young particulate and old microbial processed SOC). This simple model based on first-order kinetics was further developed by Meurer et al. (2020) to account for the interactions of soil organic matter (SOM) with soil physical properties to enable simulation of physical protection due to soil aggregation. More recently, Coucheney et al. (2024) further developed the model to account for the effects of SOC stocks on decomposition rates due to microbial energy limitation (i.e. positive and negative priming) following an approach originally proposed by Wutzler and Reichstein (2013). Compared with the original ICBM model (Andrén and Kätterer, 1997), this new model only requires two additional parameters, one to account for physical protection and one for microbial energy limitation.

Coucheney et al. (2024) introduced this simple model of SOC turnover into the new soil-crop model USSF (**U**ppsala model of **S**oil **S**tructure and **F**unction; Jarvis et al., 2024) and used it to evaluate the potential of winter wheat ideotypes with improved root system characteristics to enhance SOC stocks in a structured clay soil in Uppsala. In doing so, Coucheney et al. (2024) parameterized the SOC model from literature information, as the available site data was thought to be insufficient to unequivocally identify the model parameters. Here, we first describe the SOC model. Secondly, we present a test of model predictions and an analysis of parameter identifiability using organic C concentrations measured in the topsoil of three treatments with strongly contrasting OM inputs in a long-term field experiment in Uppsala. Finally, we perform a Monte Carlo sensitivity analysis to identify the most influential parameters in the model determining estimates of total stocks of SOC in the soil profile at steady-state. Input distributions for soil and crop parameters were defined for an agricultural production area in east-central Sweden that encompasses Uppsala. Geo-referenced data that would enable a spatially explicit test of the model for this region was not available. Instead, aggregated regional-scale soil survey data was used as a qualitative "reality-check", assuming that profiles of SOC are approximately at steady-state.

## 2    Materials and Methods

In the following, we first describe a new parsimonious model of OM turnover applicable to a single topsoil horizon, which we test using data from three contrasting cropping and fertilization treatments in the Ultuna Long-Term Soil

Organic Matter Experiment. We then derive a steady-state solution of the model and also show how it can be extended to describe OM storage and turnover in a complete soil profile. Finally, these profile-scale steady-state solutions are used to support a regional-scale sensitivity analysis and reality-check.

## 2.1 Model description

### 2.1.1 SOM turnover and storage in a single soil horizon

A dual-porosity model describing the two-way interactions between soil physical properties and SOM stocks and turnover was described by Meurer et al. (2020). In this model, SOM contents influence the total porosity and its partitioning between two pore regions in the soil (i.e. mesopores and micropores) using a simple model that describes how SOM affects aggregation. In turn, the pore size distribution determines the partitioning of root-derived inputs of OM between the two pore regions. This means that compared with a sandy soil, a larger proportion of the root OM input will enter the micropore region in a clay soil, as it predominantly consists of smaller pores. The soil pore size distribution also regulates decomposition rates with slower decomposition rates of OM stored in microporous regions of the soil. Compared with sandy soils, clay soils therefore have a greater potential for physical protection of soil C. Coucheney et al. (2024) introduced a description of the effects of microbial energy limitation according to the "*LimUptake*" variant of the model suite described by Wutzler and Reichstein (2013) into the SOM model described by Meurer et al. (2020). They also simplified the description of the transfer of SOM between the two pore regions by tillage, making the assumption that there is always a net transfer of SOM from micropore to mesopore regions. This should give more realistic simulations of the effects of tillage on SOM and also has the added benefit of allowing a straightforward solution of the model for steady-state conditions.

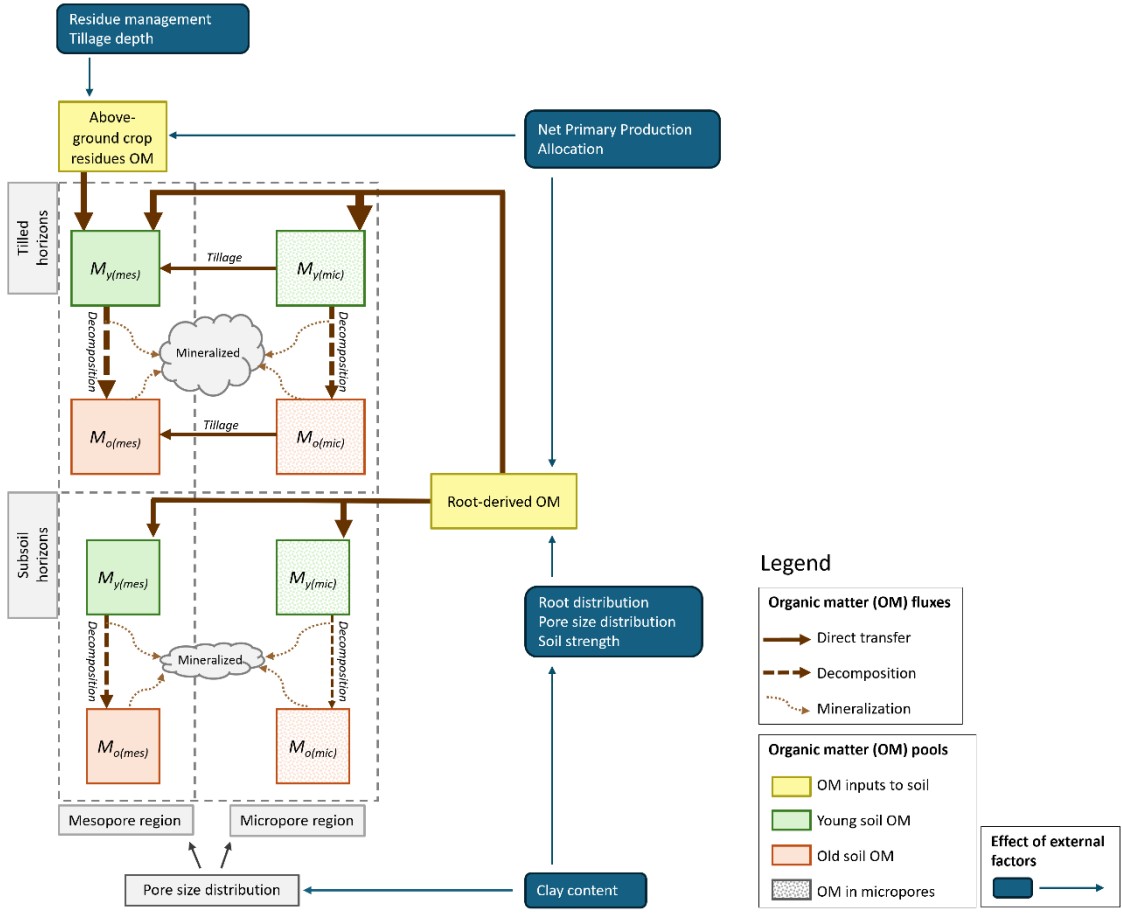

**Figure 1. Schematic diagram of the organic matter (OM) pools (M = mass, Y = young; O = old) and fluxes and the main external factors affecting OM inputs to the soil (for definitions see equations 1 to 6, 10 to 13 and 20 to 24). The grey boxes with dashed lines indicate tilled and subsoil horizons in the soil profile, both partitioned between two pore regions (micropores and mesopores). This partitioning is estimated from the soil clay content using pedotransfer functions (equations 10 to 13). OM located in the micropores is partially physically protected from decomposition (by a factor $F_p$ see equations 3 and 4). In tilled horizons, OM from the above-ground crop residues is added only to the mesopore region and a fraction of OM located in the micropores is transferred by tillage. Root-derived OM is added to both pore regions as a function of pore size distribution and soil strength through effects on root distribution in the soil profile (see equations 21-24). The sizes of the boxes and arrows illustrate that OM contents and fluxes are generally smaller in subsoil as a result of lower OM inputs, which, in turn, leads to greater energy limitation (equations 5 and 6).**

The model tracks four pools of SOM, two pools of young OM ($M_{Y(mic)}$ and $M_{Y(mes)}$) and two pools of older microbial-processed SOM ($M_{O(mic)}$ and $M_{O(mes)}$) (see Fig. 1). For both types, one part is stored in microporous regions of the soil (subscript "mic") where it is partially protected from decomposition, while the remainder is stored in regions of the soil in contact with larger mesopores (subscript "mes"), which facilitates faster decomposition (see Fig. 1). Changes in the mass of SOM in the four pools (kg m$^{-2}$) in a horizon are given by:

$$\frac{dM_{Y(mes)}}{dt} = I_a + I_r\left(1 - f_{r,mic}\right) - k_Y k_{u(mes)} M_{Y(mes)} + k_{till} M_{Y(mic)} \tag{1}$$

$$\frac{dM_{O(mes)}}{dt} = \left(\varepsilon\, k_Y k_{u(mes)} M_{Y(mes)}\right) - \left((1 - \varepsilon)\, k_O k_{u(mes)} M_{O(mes)}\right) + k_{till} M_{O(mic)} \tag{2}$$

$$\frac{dM_{Y(mic)}}{dt} = I_r f_{r,mic} - k_Y k_{u(mic)} F_p M_{Y(mic)} - k_{till} M_{Y(mic)} \tag{3}$$

$$\frac{dM_{O(mic)}}{dt} = \left(\varepsilon\, k_Y k_{u(mic)} F_p M_{Y(mic)}\right) - \left((1 - \varepsilon)\, k_O k_{u(mic)} F_p M_{O(mic)}\right) - k_{till} M_{O(mic)} \tag{4}$$

where $I_a$ and $I_r$ (kg m$^{-2}$ yr$^{-1}$) are the supply of OM from above-ground residues and roots respectively, $f_{r,mic}$ (-) is the proportion of the root-derived OM added to the micropore region, $\varepsilon$ (-) is the SOM retention coefficient, $k_Y$ and $k_O$ (yr$^{-1}$) are reference rate constants for the decomposition of young and old SOM. $k_{till}$ (yr$^{-1}$) is a rate constant regulating the transfer of SOM between pore regions by tillage, $F_p$ (-) is a factor varying from zero to unity that reduces OM decomposition rates in the micropore region to account for physical protection and $k_{u(mes)}$ and $k_{u(mic)}$ (-) are microbial energy limitation factors given by the simple model described by Wutzler and Reichstein (2013), which they derived from a simplified steady-state solution of a microbial growth model:

$$k_{u(mes)} = max\left\{0; \left(1 - \frac{A_a}{\varepsilon\left(k_Y\left(\frac{M_{Y(mes)}}{\Delta z}\right) + k_o\left(\frac{M_{o(mes)}}{\Delta z}\right)\right)}\right)\right\} \tag{5}$$

$$k_{u(mic)} = max\left\{0; \left(1 - \frac{A_a}{\varepsilon F_p\left(k_Y\left(\frac{M_{Y(mic)}}{\Delta z}\right) + k_o\left(\frac{M_{o(mic)}}{\Delta z}\right)\right)}\right)\right\} \tag{6}$$

where $A_a$ (kg m$^{-3}$ yr$^{-1}$) is a composite microbial parameter that represents a minimum C uptake flux that can support an active microbial biomass and $\Delta z$ is the horizon thickness (m). It can be seen from equations 1 and 3 that ploughed-down above-ground crop residues are presumed to lack physical protection, being incorporated into the young OM pool in contact with the larger mesopores. In contrast, some roots will grow through microporous soil regions, thereby supplying OM to the young pool on root death, as well as by root exudation.

Soil bulk density, $\gamma_b$ (kg m$^{-3}$) and OM content $f_{som}$ (kg kg$^{-1}$) are calculated from the stocks of OM as inter-linked variables (Meurer et al., 2020):

$$\gamma_b = \frac{M_{tot} + \left(\Delta z_{min} \gamma_m (1 - \phi_{min})\right)}{\Delta z} \tag{7}$$

$$f_{som} = \frac{M_{tot}}{\Delta z\, \gamma_b} \tag{8}$$

where $M_{tot}$ (kg m$^{-2}$) is the total OM stock ($= M_{Y(mes)} + M_{O(mes)} + M_{Y(mic)} + M_{O(mic)}$), $\gamma_m$ (kg m$^{-3}$) is the density of mineral matter in soil and $\phi_{min}$ is the textural porosity in soil (m$^3$ m$^{-3}$). The horizon thickness in equations 5 to 8 varies due to soil aggregation (Meurer et al., 2020):

$$\Delta z = \Delta z_{min} + \left\{ (1 + f_{agg}) \left( \frac{M_{tot}}{\gamma_o} \right) \right\} \qquad (9)$$

where $f_{agg}$ (m³ m⁻³) is the aggregation factor, $\gamma_o$ (kg m⁻³) is the density of SOM and $\Delta z_{min}$ (m) is the minimum layer thickness in a soil without SOM and aggregation porosity.

Meurer et al. (2020) equated $f_{r,mic}$ in equations 1 and 3 with the micropore fraction of the soil pore space, which varied with changes in OM stocks in each pore region. Here, in order to derive a solution for OM stocks at steady-state (see "Steady-state solution for SOM stocks"), the fraction of the root-derived OM added to the micropore region ($f_{r,mic}$ in equations 1 and 3) is assumed to be a constant and is calculated from a micropore fraction of the pore space $f_{mic}$ (-) estimated from the soil clay content, weighted by a dimensionless constant $w$ ($0 \leq w \leq 1$) to account for the effects of soil strength on the distribution of roots between the two pore regions. Using a power law function for the pore size distribution gives:

$$f_{r,mic} = w\, f_{mic} = w \left( \frac{\psi_{ae}}{\psi_{mic}} \right)^{\lambda} \qquad (10)$$

where $\psi_{ae}$ and $\psi_{mic}$ are the air-entry pressure head (m) and the pressure head (m) equivalent to the largest micropore in the soil respectively and $\lambda$ (-) is the pore size distribution index (Brooks and Corey, 1964), which is here estimated from soil clay content $f_{clay}$ (kg kg⁻¹) using the pedotransfer functions for field capacity $\theta_{fc}$ and wilting point $\theta_w$ (m³ m⁻³) derived from a database of water retention curves for Swedish agricultural soils by Kätterer et al. (2006):

$$\lambda = \frac{log\left( \frac{\theta_w}{\theta_{fc}} \right)}{log\left( \frac{0.5}{150} \right)} \qquad (11)$$

$$\theta_{fc} = 0.27 + 0.325\, f_{clay} \qquad (12)$$

$$\theta_w = 0.004 + 0.5 f_{clay} \qquad (13)$$

Thus, in this simpler version of the model described by Meurer et al. (2020), changes in SOM contents affect the porosity and bulk density but not the pore size distribution.

### 2.1.2 Steady-state solution for SOM stocks

From equations 1 to 4, steady-state SOM stocks in the four pools are given as:

$$M_{Y(mic)} = \left( \frac{I_r\, f_{r,mic}}{\{k_Y F_p k_{u,mic}\} + k_{till}} \right) \qquad (14)$$

$$M_{Y(mes)} = \left( \frac{I_a + I_r(1 - f_{r,mic}) + \{k_{till} M_{Y(mic)}\}}{k_Y k_{u,mes}} \right) \qquad (15)$$

$$M_{O(mic)} = \left( \frac{\varepsilon k_Y k_{u,mic} F_p M_{Y(mic)}}{\{(1-\varepsilon)k_O F_p k_{u,mic}\} + k_{till}} \right) \qquad (16)$$

$$M_{O(mes)} = \left( \frac{\{\varepsilon k_Y k_{u,mes} M_{Y(mes)}\} + k_{till} M_{O,mic}}{(1-\varepsilon)k_O k_{u,mes}} \right) \qquad (17)$$

Equations 14 to 17 show that the steady-state stocks depend on $k_u$, while $k_u$, in turn, depends on the stocks (equations 5 and 6). An iterative procedure is first used to derive a value of $k_{u(mic)}$ at steady-state that simultaneously satisfies equations 6, 14 and 16. The steady-state stocks in the mesopore region (equations 15 and 17) depend on

the value of $k_{u,mes}$ at steady-state. This can now be calculated directly by substituting equations 15 and 17 into equation 5:


$$k_{u,mes} = \frac{1}{1 + \left\{\dfrac{A_a\,\Delta z}{\varepsilon\left(i^* + \left(\dfrac{\varepsilon i^* + k_{till}M_{O(mic)}}{1-\varepsilon}\right)\right)}\right\}} \qquad (18)$$

where $i^*$ is the input of OM to the mesopore region given by:

$$i^* = I_a + I_r\left(1 - f_{r,mic}\right) + k_{till}M_{Y(mic)} \qquad (19)$$

### 2.1.3 Application of the model to a soil profile

The model can be applied to a soil profile consisting of two or more soil horizons by expressing $k_{till}$, $I_a$, $I_r$, and $w$

as a function of soil depth, keeping all the other parameters constant. For the sake of simplicity, the textural porosity $\phi_{min}$ (equation 7) could also vary with depth in the soil, but it is assumed to take a constant value in the following,. Tillage is here assumed to affect SOM turnover only in the uppermost horizon, with $k_{till}$ set to zero for all other horizons. Above-ground crop residues $I_a$ are given by:

$$I_a = Y\left(\frac{1}{HI} - 1\right) f_{inc} \qquad (20)$$

where $Y$ is the yield (kg m$^{-2}$), $HI$ (-) is the harvest index (the ratio of yield to total above-ground biomass) and $f_{inc}$ is the proportion of the above-ground residues incorporated into soil. The partitioning of $I_a$ among the soil horizons can be defined by the user, but should reflect tillage systems and depths of cultivation. The total input of root-derived OM, $I_r$ is given by:

$$I_{r(tot)} = \frac{Y f_{bg}}{HI(1 - f_{bg})} \qquad (21)$$

where $f_{bg}$ is the proportion of net primary production that is allocated below-ground, including both root growth and exudates. Root-derived OM is added to the soil horizons in the profile according to a two-parameter logistic function, which represents the distribution of roots with depth in the soil (e.g. Schenk and Jackson, 2002; Fan et al., 2016):

$$P = \frac{1}{1 + \left(\dfrac{z}{D_{50}}\right)^c} \qquad (22)$$

where $P$ is the fraction of the total root biomass found above a depth $z$, representing the lower boundary of the horizon in question, $c$ is a root distribution parameter and $D_{50}$ is the depth above which 50% of the root biomass is recovered, which is given by:

$$D_{50} = \frac{D_{95}}{\left(\frac{1}{0.95} - 1\right)^{\frac{1}{c}}} \qquad (23)$$

where $D_{95}$ is the depth (m) above which 95% of the total root biomass is recovered. With this function, a small

fraction of the root biomass is found below the depth of the soil profile. This additional fraction of the root biomass is added to the upper two horizons in equal amounts.

Finally, the weighting function to account for the effects of soil strength on the distribution of roots between the two pore regions is given by:

$$w = EXP\big(-w_s(z - z_1)\big) \tag{24}$$

where the constant $w_s$ (m$^{-1}$) reflects the effects of increasing soil strength with depth on the distribution of roots between soil micropore and mesopore regions and $z_1$ is the depth to the lower boundary of the uppermost soil horizon. It can be seen from equation 24 that $w = 1$ for the uppermost horizon, so that the root-derived OM in this layer is partitioned between the pore regions directly proportional to their estimated respective partial volumes.

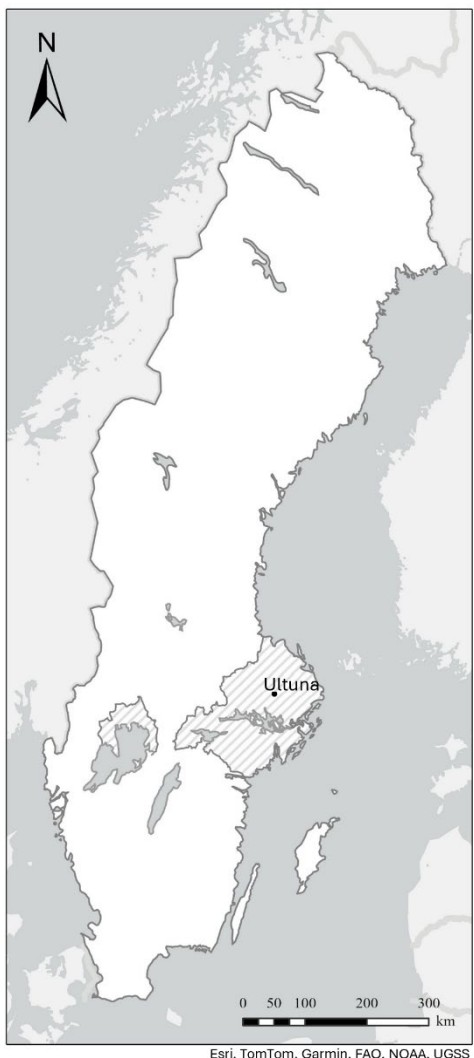

**Figure 2. Map of Sweden (in white) showing the location of the Ultuna Long-term Soil Organic Matter**
**Experiment (Uppsala, Sweden) and the extent of the production area PO4 (shaded area in grey). Drawn by**
**Anna Lindahl, SLU, from Esri, TomTom, Garmin, FAO, NOAA, and USGS**

**2.2    Model applications**

**2.2.1    Long-term transient simulations of SOC under contrasting cropping and fertilization**

We performed a test of the model described by equations 1 to 13 using data from the Ultuna Long-Term Soil
Organic Matter Experiment located at Uppsala, east-central Sweden (59.8° N, 17.7° E; Fig. 2; Pold et al., 2025).

The mean annual temperature at Ultuna is 7 °C and the mean annual precipitation is 570 mm. The texture in the uppermost 20 cm of soil is clay loam (37 % clay, 41 % silt and 22 % sand). In this study, we make use of SOC contents measured in the topsoil (0-20 cm depth) from the start of the trial in 1956 until 2010 in three treatments with contrasting inputs of organic matter: an uncropped fallow treatment (*"Fallow"*) and two cropped treatments ("N fertilized" and "N fertilized + straw"), both of which are supplied with $Ca(NO_3)_2$ every year at the time of sowing at a rate of 80 kg N $ha^{-1}$ $year^{-1}$. Most (ca. 95%) of the above-ground crop residues are removed at harvest in autumn and straw is applied biennially to the treatment "N fertilized + straw" after harvest at an equivalent annual rate of 4.2 t $ha^{-1}$. Maize (*Zea mays*) has been grown on the cropped plots since 2000. Before 2000, the crop rotation included barley (*Hordeum vulgare*), oats (*Avena sativa*), beets (*Beta vulgaris*) (prior to 1967) and rape (*Brassica napus*). All the plots are dug by hand after harvest each year to a depth of 20 cm to simulate ploughing as the plots are too small (4 $m^2$) to be managed in the same way as a farmer's field. We refer readers to Persson and Kirchmann (1994) and Kätterer et al. (2011) for more details of the design of the field experiment.

Inputs of OM from above-ground crop residues and root-derived OM were estimated following Kätterer et al. (2011), who made use of the allocation functions dependent on crop yields derived by Bolinder et al. (2007), together with a Michaelis-Menten function to estimate the proportion of the root-derived OM that was presumed to have been input to the topsoil (0-20 cm). Here, we simplified this method by using average OM inputs in each treatment for the experimental period (1956-2010) based on annual values calculated for the different crops in the rotation.

The model was simultaneously calibrated to the measurements of total SOC from the three treatments using the Generalized Likelihood Uncertainty Estimation (GLUE) method (Beven, 2006; Beven and Binley 2014; Juston et al., 2010). This is because we wanted to critically test the model to see if it was possible to obtain acceptable parameterizations common to all three of the treatments. Inspection of the model equations led us to expect to encounter significant equifinality. Therefore, only six of the fifteen parameters were included in the GLUE analysis, with their prior uncertainty ranges shown in Table 1. The OM supply prior to the start of the experiment and the fraction of this OM supplied as straw, were included in the calibration process to help initialize the SOM pools during a common 5000-year spin-up period. The four other parameters, which were considered difficult to identify "a priori" from experimentation, but which were expected to be sensitive and therefore potentially identifiable by calibration, were treated as uncertain (Table 1). We ran 12000 simulations using Latin Hypercube Sampling to sample uniform distributions between the minimum and maximum values for the six uncertain parameters (Table 1). The remaining nine parameters were set to fixed values (Table 2) as they could be estimated from measurements (e.g. $f_{clay}$, $f_{agg}$, $F_p$) or they were not expected to be sensitive (e.g. $k_y$, Andrén and Kätterer, 1997; Juston et al., 2010; Meurer et al., 2020), or both (e.g. $\psi_{ae}$, $\psi_{mic}$, $\phi_{min}$, $\gamma_o$, $\gamma_m$). These fixed parameters included the soil physical properties, since an analysis of soil structure dynamics was not the main focus of this modelling study, which employs a slightly simplified description of the interactions between soil aggregation and SOM. It can be noted that the final bulk densities for the 30 best simulations (see below) derived varied between 1.2 and 1.3 g $cm^{-3}$ using the fixed parameter values shown in Table 2 (with "N fertilized + straw" < "N fertilized" < "Fallow"). This matched reasonably well the magnitude and order of the measured values reported for the three treatments in Kätterer et al. (2011), which were 1.43, 1.28 and 1.21 g $cm^{-3}$ respectively.

**Table 1. Six model parameters selected for the calibration to the Ultuna Long-Term Soil Organic Matter Experiment and their initial parameter uncertainty ranges**

| Parameter | Symbol | Units | Prior uncertainty ranges |
|---|---|---|---|
| Total OM input during spin-up | $I_a + I_r$ | kg m$^{-2}$ y$^{-1}$ | 0.25-0.45 |
| Straw fraction of OM input during spin-up | $I_a/(I_a + I_r)$ | - | 0.65-0.85 |
| Rate constant for OM transfer by tillage between pore regions | $k_{till}$ | y$^{-1}$ | 0.00- 0.01 |
| Reference decomposition rate constant for old OM | $k_o$ | y$^{-1}$ | 0.06- 0.10 |
| OM retention coefficient | $\varepsilon$ | - | 0.20- 0.45 |
| Microbial energy limitation factor | $A_a$ | kg m$^{-3}$ y$^{-1}$ | 0.10- 0.30 |

**Table 2. Nine model parameters fixed at constant values during the calibration based on field measurements at Ultuna or literature data**

| Parameter | Symbol | Units | Value | Source |
|---|---|---|---|---|
| Clay content | $f_{clay}$ | kg kg$^{-1}$ | 0.36 | Persson and Kirchmann (1994); Pold et al. (2025) |
| Density of organic matter | $\gamma_o$ | kg m$^{-3}$ | 1200 | Meurer et al. (2020); Coucheney et al. (2024) |
| Density of mineral matter | $\gamma_m$ | kg m$^{-3}$ | 2700 | Meurer et al. (2020); Coucheney et al. (2024) |
| Textural porosity | $\phi_{min}$ | m$^3$ m$^{-3}$ | 0.5 | Coucheney et al. (2024) |
| Aggregation factor | $f_{agg}$ | m$^3$ m$^{-3}$ | 3 | Meurer et al. (2020) |
| Physical protection factor | $F_p$ | - | 0.2 | Kravchenko et al. (2015) |
| Air-entry pressure head | $\psi_{ae}$ | m | - 0.2 | Coucheney et al. (2024) |
| Pressure head equivalent to the largest micropore in soil | $\psi_{mic}$ | m | - 6.0 | Killham et al. (1993); Strong et al. (2004); Ruamps et al. (2011) |
| Reference decomposition rate constant for young OM | $k_y$ | y$^{-1}$ | 0.8 | Andrén and Kätterer (1997) |

The model efficiency *EF* was used as the likelihood function in GLUE:

$$EF = 1 - \frac{\sum_{i=1}^{n}(O_i - P_i)^2}{\sum_{i=1}^{n}(O_i - \bar{O})^2} \tag{25}$$

where $O$ and $P$ are observed and predicted values, $\bar{O}$ is the mean of the observations and $n$ is the number of observations. The maximum value of *EF* is one, when predictions and observations are identical, while a negative value implies a poor model, since it means that taking the average of the observations would give a better prediction. For each simulation, individual model efficiencies were calculated for each treatment and the mean EF value for the three treatments was used as a metric to identify acceptable parameters sets. This was done to obtain a robust parameterization by selecting parameter sets that simultaneously fitted all three treatments well. The

number of acceptable parameter sets was determined such that the range of variation of their predictions approximately covered the variations observed in the measurements. With this criterion, 30 of the 12000 parameter sets were identified as acceptable. Note that this low acceptance rate is a consequence of the inefficient sampling inherent to the GLUE method and says nothing about the quality of the model.

### 2.2.2 Steady-state calculations: sensitivity analysis and reality-check

We performed a Monte Carlo sensitivity and uncertainty analysis to assess the relative importance of the model parameters for predictions of the steady-state stocks of SOM in the soil profile (equations 7 to 24; Table 3). The analysis was based, to the extent possible, on data and information available for the Ultuna field site as well as soil survey and cropping data (e.g. crop yields, soil clay content) for the agricultural production area number 4 in east-central Sweden (i.e. the region in which Ultuna is located; Fig. 2). Literature information was used to determine parameter distributions in the absence of data at the local or regional scale (Table 3). Of all the model parameters, only $\psi_{ae}$ was fixed at a constant value, as there is no 'a priori' physical reason to expect that its value should vary among different soils. We assumed normal distributions when the data was considered sufficient to support such a distribution. Uniform distributions were used otherwise (Table 3). One thousand parameter sets were generated from these distributions by random sampling.

Calculations were performed for a soil profile 120 cm in depth, divided into four soil horizons (0-20, 20-40, 40-60 and 60-120 cm). We added 80% of the above-ground residues $I_a$ (equation 20) to the uppermost horizon in the soil profile and the remaining 20% to the horizon below. For all 1000 parameter sets, we calculated the SOM stock in each horizon and in the whole soil profile at steady-state. For each soil horizon, we also calculated the steady-state bulk density and SOM contents as well as the mean residence time of SOM as the steady-state SOM stock divided by the input/output flux.

We used a multiple linear regression model to characterize variations in the steady-state SOM stocks in the profile ($y$), such that the normalized coefficients ($\beta_1, \beta_2 \ldots \beta_n$) can be used as a metric of sensitivity to variation in the parameters ($x_1, x_2 \ldots x_n$) (Saltelli and Annoni, 2010):

$$y = \beta_0 + \beta_1 x_1 + \beta_2 x_2 + \ldots \beta_n x_n \tag{26}$$

Aggregated data for SOC contents measured at three depth intervals (0-20, 20-40 and 40-60 cm depth) for soils in production area number 4 (Figure 2; n = 611, 100 and 100 respectively) were extracted from the national soil and crop inventory carried out from 2001 to 2007 (Eriksson et al., 2010) and used as a qualitative "reality-check" for the model calculations. Note that, as a consequence of simulating links to soil physical properties, the model calculates SOM contents, whereas SOC was measured. In converting from one to the other, we assumed that organic C constituted 50% of the SOM. Likewise, calculated bulk densities at zero to 20 cm and 40 to 60 cm depth were compared with data available for soil profiles (n = 54) located in production area 4 (Klöffel et al., 2024). The model parameters required to convert calculated SOM stocks to estimates of SOM contents using equations 7 to 9 were set to the fixed values used in the model calibration (Table 2), with the exception of the textural porosity which was reduced from 0.5 to 0.4, as the latter value was considered to be more representative for most soils (Klöffel et al., 2024). Note that the textural porosity was also assumed to be constant with depth in the soil.

**Table 3. Parameter input distributions in the sensitivity analysis. In the case of uniform distributions, minimal and maximal values are shown (Min.; Max.) while in the case of normal distribution the mean and standard deviation are shown (Mean; St. dev.).**

| Group | Parameters (symbol, unit) | Distribution | Source |
|---|---|---|---|
| Crop growth and residue inputs | Yield ($Y$, kg m$^{-2}$) | Normal (0.50; 0.05) | SCB, Statistics Sweden: https://www.statistikdatabasen.scb.se/pxweb/en/ssd/START__JO__JO0601/SkordarL2/ |
| | Harvest index ($HI$, -) | Normal (0.40; 0.05) | Hay (1995); Kätterer et al. (1997); Coucheney et al. (2024) |
| | Fraction of net primary production allocated belowground ($f_{bg}$, -) | Normal (0.200; 0.025) | Bolinder et al. (2007); Kätterer et al. (2011) |
| | Fraction of aboveground crop residues incorporated ($f_{inc}$, -) | Normal (0.65; 0.10) | Smerald et al. (2023) |
| | Root depth ($D_{95}$, m) | Uniform (0.8; 1.2) | Jackson et al. (1996); Kätterer et al. (2011); Fan et al. (2016) |
| | Root distribution factor (c, -) | Uniform (-1.2; -0.9) | Fan et al. (2016) |
| Tillage | Rate constant for OM transfer between pore regions ($k_{till}$, y$^{-1}$) | Uniform (0.000; 0.006) | This study |
| Organic matter turnover | Reference decomposition rate constant for young organic matter ($k_Y$, y$^{-1}$) | Uniform (0.6; 1.0) | Andrén and Kätterer (1997) |
| | Reference decomposition rate constant for old organic matter ($k_O$, y$^{-1}$) | Uniform (0.06; 0.10) | This study |
| | OM retention coefficient (ε, -) | Uniform (0.30; 0.35) | This study |
| | Physical protection factor ($F_p$, -) | Uniform (0.1; 0.3) | Kravchenko et al. (2015) |
| | Microbial energy limitation factor ($A_a$, kg m$^{-3}$ y$^{-1}$) | Uniform (0.1; 0.3) | This study |
| Soil physical properties | Clay content ($f_{clay}$, kg kg$^{-1}$) | Normal (0.3; 0.1) | Eriksson et al. (2010) |
| | Factor for soil strength effects on root distribution between pore regions ($w_s$, m$^{-1}$) | Uniform (2; 4) | This study |
| | Pressure head defining the largest micropore ($\psi_{mic}$, m) | Uniform (-30; -6) | Killham et al. (1993); Strong et al. (2004); Ruamps et al. (2011) |

## 3 Results and Discussion

### 3.1 Long-term transient simulations

Figure 3 shows that the model could be calibrated to match simultaneously the changes in SOC contents measured in the three treatments at the Ultuna Long-Term Soil Organic Matter Experiment during the 50 year period, with the spread of the simulations from the 30 best parameter sets approximately matching the observed variation in SOC among the four replicate plots. Table 4 shows simulated SOM balances for the three treatments. The total input of crop residues in the "N fertilized + straw" treatment is roughly three times that of the "N-fertilized" treatment without straw addition. The calculated inputs of OM derived from roots were similar (Table 4), so that larger inputs of straw accounted for almost all of the difference in OM inputs between these two treatments. However, according to the simulations, almost 88% of the additional OM input in the "N fertilized + straw" treatment was lost as a consequence of enhanced mineralization, with only 12% remaining in the soil. While above-ground crop residues are thought to be less persistent in soil than root-derived residues, the relative importance of several potential underlying mechanisms that could explain this finding is still unclear (e.g. Rasse et al., 2005; Kätterer et al., 2011). It can be noted here that the model does not consider any differences in the quality of root- and straw-derived OM. Instead, the model suggests that the comparatively small difference in OM stocks at the end of the experiment in the two treatments in relation to the large difference in OM inputs is a result of two processes: firstly, straw incorporated in the "N fertilized + straw" treatment is solely added to the mesopore region, which does not afford any physical protection. In contrast, a certain proportion, $f_{mic}$, of root-derived OM is added to the physically-protected micropore region. Secondly, mineralization rates in the "N-fertilized" treatment without straw addition are reduced by microbial energy limitation as a consequence of an overall decrease in OM stocks due to the near total removal of above-ground crop residues. Taking both these processes into account (physical protection and microbial energy limitation; see equations 1 to 6) enabled the model to reproduce the time-courses of SOC contents in the two treatments with identical parameterizations. Models that do not consider these processes would need to adjust their parameter values (either decomposition rate constants or the retention coefficient) to match this data (e.g. Poeplau et al., 2015).

Figure 4 shows that only one of the parameters included in the calibration procedure (the OM retention coefficient, ε) was well constrained by the data, with acceptable values lying within a narrow range (ca. 0.30 to 0.35). In contrast, for the other five parameters, simulations with large model efficiencies could be found across almost the entire prior uncertainty ranges (Fig. 4). An inspection of the mathematical structure of the model suggests that such a high degree of equifinality should be expected, as many of the key parameters should be strongly correlated (Coucheney et al., 2024). For the 30 best parameter sets, Fig. 5 demonstrates that this is indeed the case for the four parameters regulating decomposition rates in the model (ε, $k_o$, $A_a$ and $k_{till}$). These strong correlations of $k_o$, $A_a$ and $k_{till}$ with ε mean that, in practice, all four parameters are well constrained by the calibration. The acceptable ranges for these four parameters shown in Fig. 5 were utilized in the sensitivity analysis (Table 3).

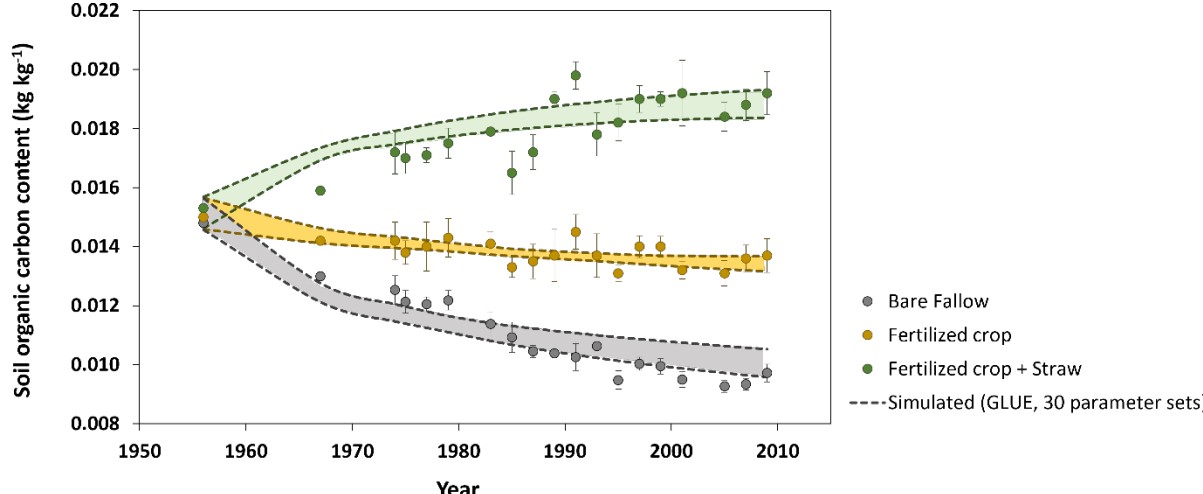

**Figure 3. Comparisons of measured SOC contents (symbols are the means of four replicates and the bars are standard deviations) with the 30 best simulations from the GLUE analysis (the dashed lines indicate ranges)**

**Table 4. Simulated mass balances (kg m$^{-2}$) for SOM for the 55-year experimental period (1956 to 2010) at the Ultuna Long-Term Soil Organic Matter Experiment. Values shown for mineralization are the means and standard deviations (in brackets) for the 30 best simulations. Values for change of stocks in brackets are the percentage changes in relation to the original stock of SOM.**

| | Treatment | | |
|---|---|---|---|
| Component | Fallow | N fertilized | N fertilized + straw |
| Inputs: | | | |
| - Below-ground residues[1] | 0.44 | 9.85 | 10.67 |
| - Above-ground residues | 0.00 | 1.82 | 22.94 |
| - Total crop residue input | 0.44 | 11.67 | 33.61 |
| Outputs: | | | |
| - Mineralization in soil | 3.01 | 12.53 | 31.75 |
| | (0.18) | (0.16) | (0.20) |
| Change of SOM stock: | -2.57 | -0.86 | 1.86 |
| | (-35.1%) | (-11.7%) | (+25.4%) |

[1] estimated using the algorithms presented by Bolinder et al. (2007) and Kätterer et al. (2011)

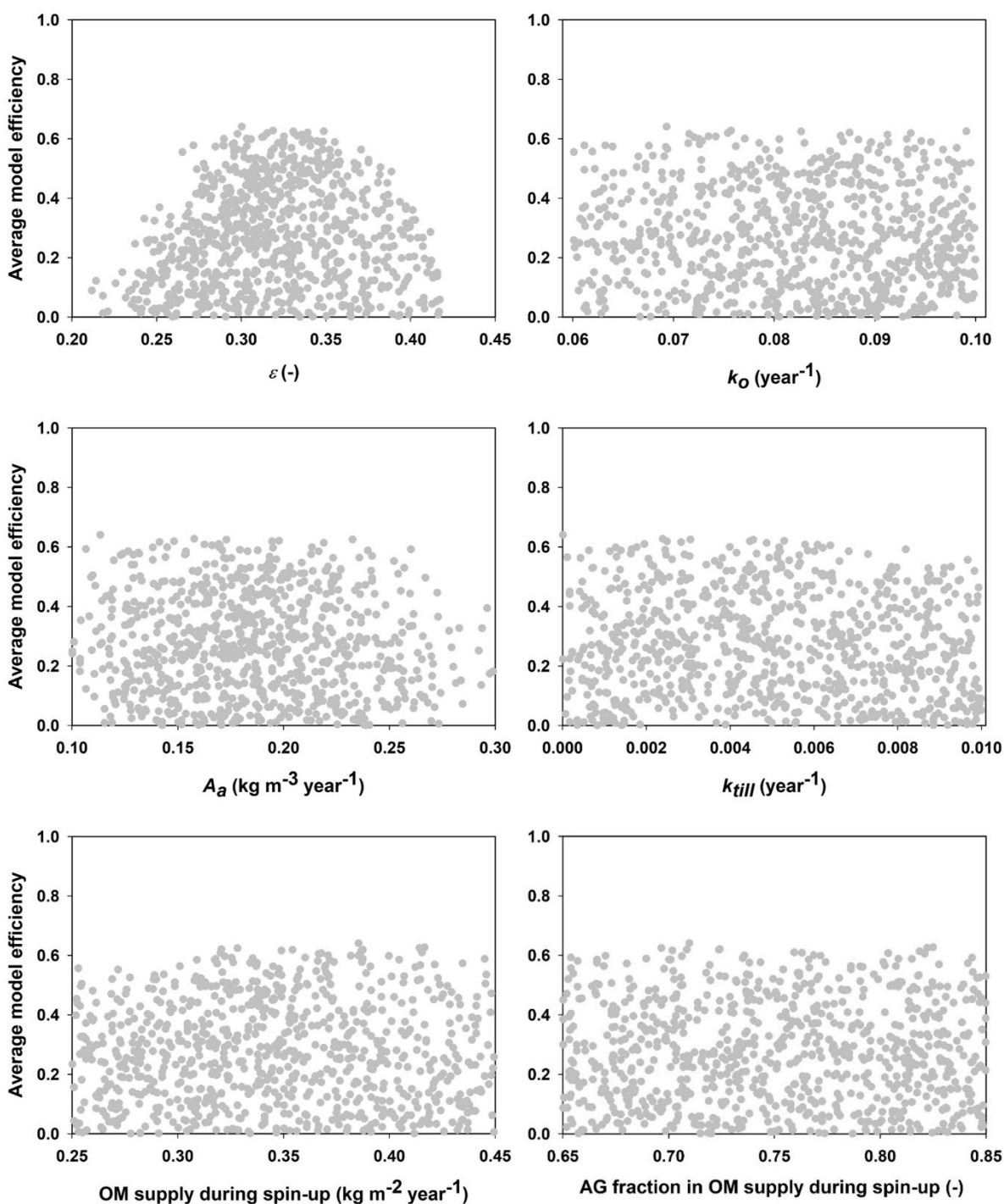

**Figure 4. Mean model efficiencies for each parameter set (only simulations with model efficiencies larger than zero are shown) plotted against the values for the six parameters in the GLUE analysis (refer to table 1 for parameter definitions and descriptions; OM = organic matter, AG = above-ground).**

### 3.2 Steady-state calculations

365   A qualitative comparison with soil survey data for agricultural land in east-central Sweden (production area number 4) suggests that despite its simplicity the model estimates of steady-state SOC and bulk density in the soil

profile lie mostly within the range of variation encountered in the region (Fig. 6; Fig. A1). Nevertheless, quantile-quantile plots show that the distributions of simulated and measured values of SOC and bulk density are different; especially at the tails, due to the much larger spread in the measurements compared with the calculations and especially the occurrence of a number of outliers with large values of organic carbon contents and small values of bulk density. This is not surprising because the calculations do not include the effects of all factors affecting SOC and bulk density. The large values of SOC content (and small values of bulk density) almost certainly correspond to locations in the region with wet soils due to topography (i.e. flood plains, depressions). The model, as it is formulated here, does not include the effects of excess soil moisture on decomposition rates. As a further qualitative "reality check", Fig. 7 shows distributions of the mean residence times of SOM calculated for the four horizons in the soil profile. Median values (ca. 20 years) and distributions of residence times estimated for the topsoil are similar to those estimated by Poeplau et al. (2021) for German agricultural soils, and they also lie at the high end of the range in the global analysis reported by Chen et al. (2020) for croplands (mean = 9.5 years, standard deviation = 6 years, n = 217). Taken together with Fig. 6, this gives us confidence that the results of the sensitivity analysis presented in the following should be reasonably well grounded in reality. As also shown by Coucheney et al. (2024), the model simulates much longer mean residence times in subsoil horizons, due to microbial energy limitation and physical protection, with median values of ca. 300 years (Fig. 7). These model estimates of mean OM residence times in the subsoil are also similar to the median age of soil organic carbon estimated from isotope data in the global analysis of Balesdent et al. (2018) for tropical forests and grasslands.

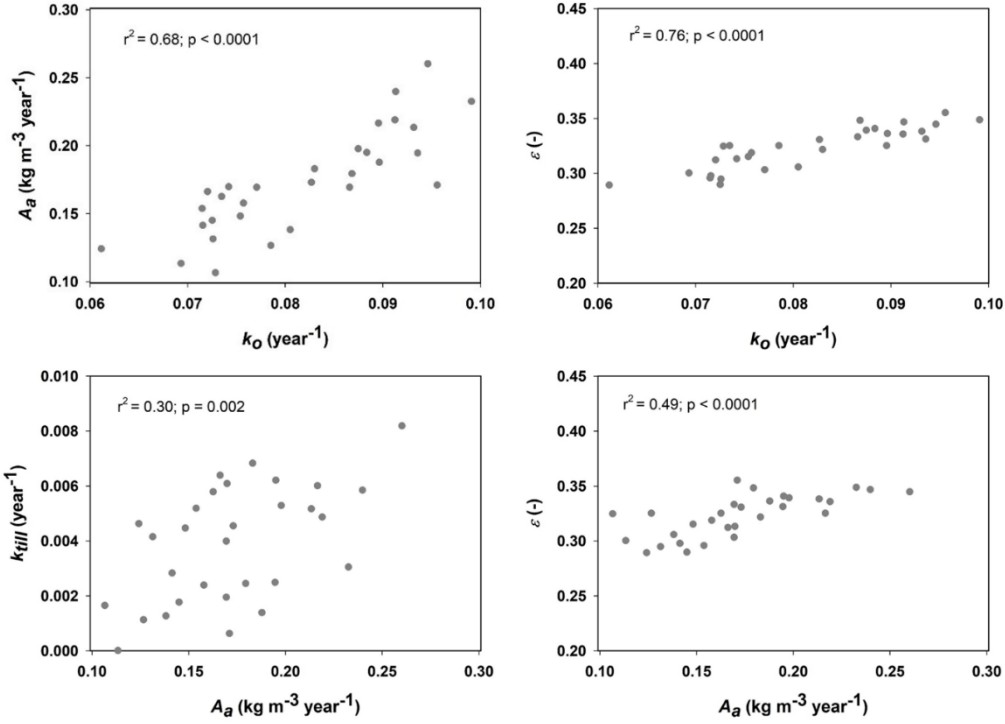

**Figure 5. Inter-relationships among four of the six model parameters included in the calibration procedure ($A_a$ is the microbial energy limitation factor, $k_o$ is the reference rate constant for decomposition of old OM, $\varepsilon$ is the OM retention co-efficient and $k_{till}$ is the rate constant for OM transfer by tillage between pore regions). Relationships are shown for the 30 best parameter sets identified in the GLUE analysis (refer to table 1 for parameter definitions and descriptions).**

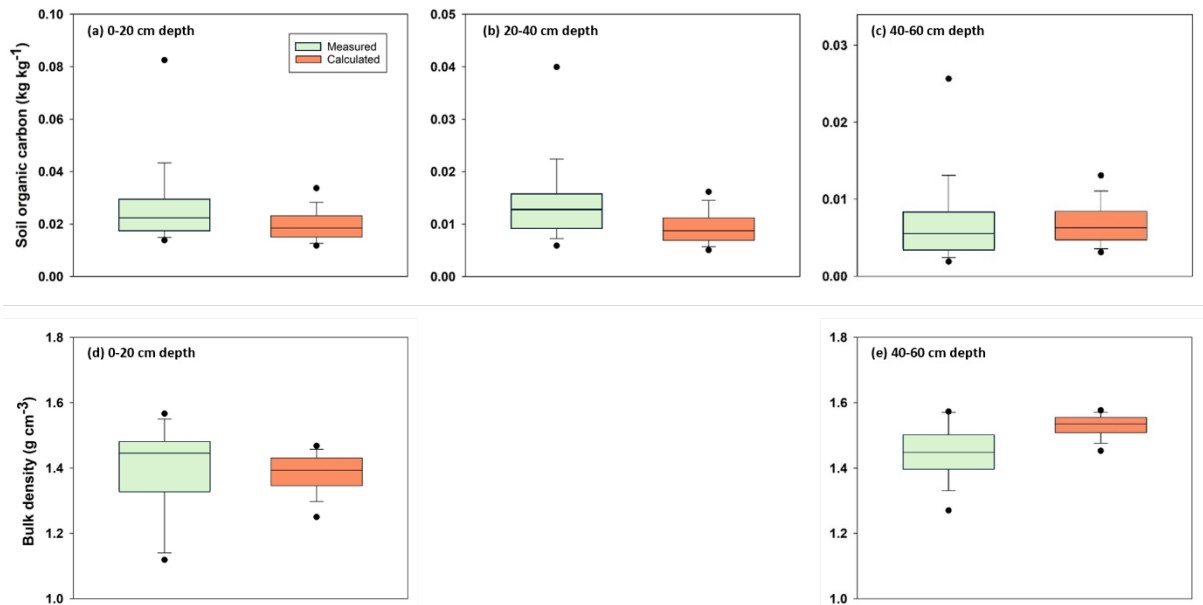

**Figure 6. Comparison of the distributions of SOC contents (a to c) and soil bulk density (d and e) measured at three and two depths respectively, for soil profiles located in east-central Sweden (production area number 4; Eriksson et al., 2010) with distributions calculated in the model sensitivity analysis. Horizontal lines show median values, the box defines the inter-quartile range, error bars define 10th and 90th percentiles and solid symbols indicate 5th and 95th percentiles. Note the differences in the y-axis scales for soil organic carbon contents.**

Table 5 shows that the most sensitive parameters in the model are those determining decomposition rates of SOM, especially the rate constant for microbial-processed OM, $k_o$, the parameter regulating microbial energy limitation, $A_a$, and the parameter regulating the degree of physical protection of OM stored in micropores, $F_p$. The soil clay content, which together with $F_p$, determines the extent to which physical protection is expressed in soils of contrasting texture, is also a relatively sensitive model parameter (Table 5). Along with the OM retention coefficient, $\varepsilon$, the three parameters determining inputs of above-ground crop residues (i.e. the fraction incorporated, $f_{inc}$, and crop yields and harvest index) also exert a strong control on SOM stocks in the soil profile (Table 5). The results of the sensitivity analysis also illustrate the importance of below-ground production for soil profile C stocks calculated by the model (parameter $f_{bg}$, fraction of NPP allocated below-ground; Table 5), reflecting the assumptions in the model concerning the greater persistence of root-derived OM discussed earlier. An increase of 25% in the fraction of NPP allocated to roots, $f_{bg}$, increases steady-state SOM stocks by ca. 8%. Transient simulations run with the USSF model for winter wheat grown on Ultuna clay soil presented by Coucheney et al. (2024) illustrate what might be achievable in a shorter 30-year time perspective in the context of climate change mitigation: for the same 25% increase in below-ground C allocation, the USSF model simulated increases in C stocks of ca. 1.4%. In contrast to below-ground production, the sensitivity analysis suggests that root depth and distribution would have little impact on soil profile stocks of OM (Table 5). However, in comparison with soil-crop models such as USSF, the limitations of the simpler model described here should be borne in mind,

in particular the lack of any feedback between root system development and crop growth, and thus residue production. In reality, root depth and distribution may play a larger role for soil C stocks. Thus, the transient simulations performed with the full USSF soil-crop model for winter wheat on Ultuna clay soil by Coucheney et al. (2024) suggested that deeper rooting would increase water uptake and crop growth in dry summers, leading to 3-5% increases in SOM stocks in a 30-year perspective. Table 5 suggests that tillage is one of the least sensitive factors affecting SOM stocks at steady-state: doubling the tillage intensity parameter in the model, $k_{till}$, only reduces SOM stocks by 4 to 5%. It must be admitted, however, that the simple description of tillage effects in the model is yet to be rigorously and systematically tested. Nevertheless, in a meta-analysis of long-term experiments in boreal/temperate climates, Haddaway et al. (2017) and Meurer et al. (2018) found larger SOC stocks under no-till compared with conventional tillage in the topsoil, but no significant differences in total SOC stocks in these two tillage systems in soil profiles to 60 cm depth.

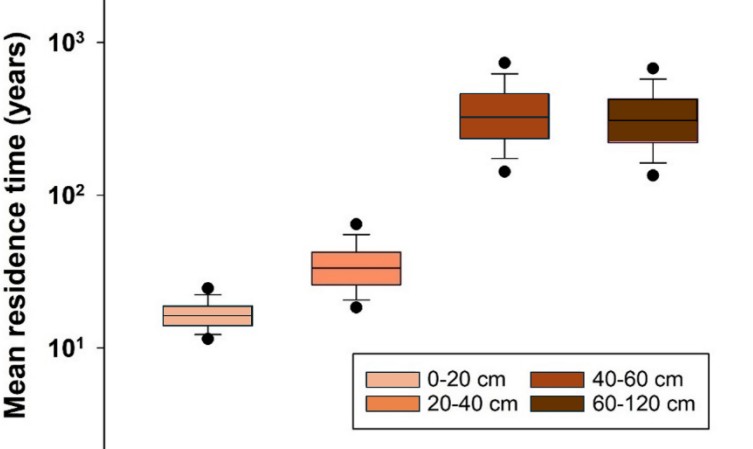

**Figure 7. Distributions of mean residence times for SOM calculated in the sensitivity analysis for four depths in the soil profiles of production area 4 in east-central Sweden. Horizontal lines show median values, the box defines the inter-quartile range, error bars define 10th and 90th percentiles and solid symbols indicate 5th and 95th percentiles.**

**Table 5. Parameter sensitivity (($\beta_i$ = normalized regression coefficients, see equation 26)**

| Parameter | | $\beta_i$ |
|---|---|---|
| $k_o$ | Decomposition rate constant (old OM) | -0.833 |
| $F_p$ | Physical protection factor | -0.695 |
| $A_a$ | Microbial energy limitation factor | 0.606 |
| $HI$ | Harvest index | -0.513 |
| $Y$ | Crop Yield | 0.401 |
| $\varepsilon$ | OM retention coefficient | 0.329 |
| $f_{bg}$ | Fraction of NPP allocated below-ground | 0.291 |
| $k_y$ | Decomposition rate constant (young OM) | -0.174 |
| $f_{clay}$ | Clay content | 0.128 |
| $f_{inc}$ | Fraction of above-ground residues incorporated | 0.127 |
| $k_{till}$ | Tillage transfer coefficient | -0.045 |
| $w_s$ | Factor for soil strength effects on root distribution | 0.035 |
| $D_{95}$ | Root depth | -0.023 |
| $\psi_{mic}$ | Pressure head defining micropore region | -0.015 |
| $c$ | Root depth distribution factor | -0.009 |

## 4    Conclusions

We presented here a novel parsimonious or "minimalist" model that simulates the emergent effects of soil texture and soil structure on C stocks and turnover rates in soil profiles by mimicking two of the key processes involved in C stabilization (i.e. physical protection and microbial energy limitation). Parameters controlling these processes were also found to be among the most sensitive in the model. However, the decomposition rate constant for old microbial-processed OM, $k_o$ was the most sensitive parameter in the model. Although $k_o$ should be considered as a lumped parameter reflecting the influence of various processes, the available experimental evidence suggests that the strength of adsorption and OM-mineral interactions controlling the bioavailability of the substrate (i.e. chemical protection) should be the most important factor underlying its variation (e.g. Lehmann and Kleber, 2015; Mathieu et al., 2015; Doetterl et al., 2015). The development of pedotransfer approaches (van Looy et al., 2017) to estimate $k_o$ using soil properties such as clay content and clay mineralogy, pH and Al and Fe oxides (e.g. Mathieu et al., 2015; Rasmussen et al., 2018; Fukumasu et al., 2021) would therefore be helpful in supporting predictive model applications at larger scales.

The comparisons of model simulations with local- and regional-scale data confirm that it shows promise. Despite equifinality, the parameters regulating decomposition in the model could be identified within reasonably narrow ranges using data from a long-term field experiment with three treatments characterized by strongly contrasting OM inputs for more than 50 years. Ideally, the model should now be further tested at multiple sites using data from long-term field experiments, including comparisons of alternative cropping systems and tillage management (i.e. no-till vs. conventional systems).

**Appendices**

Appendix A1:

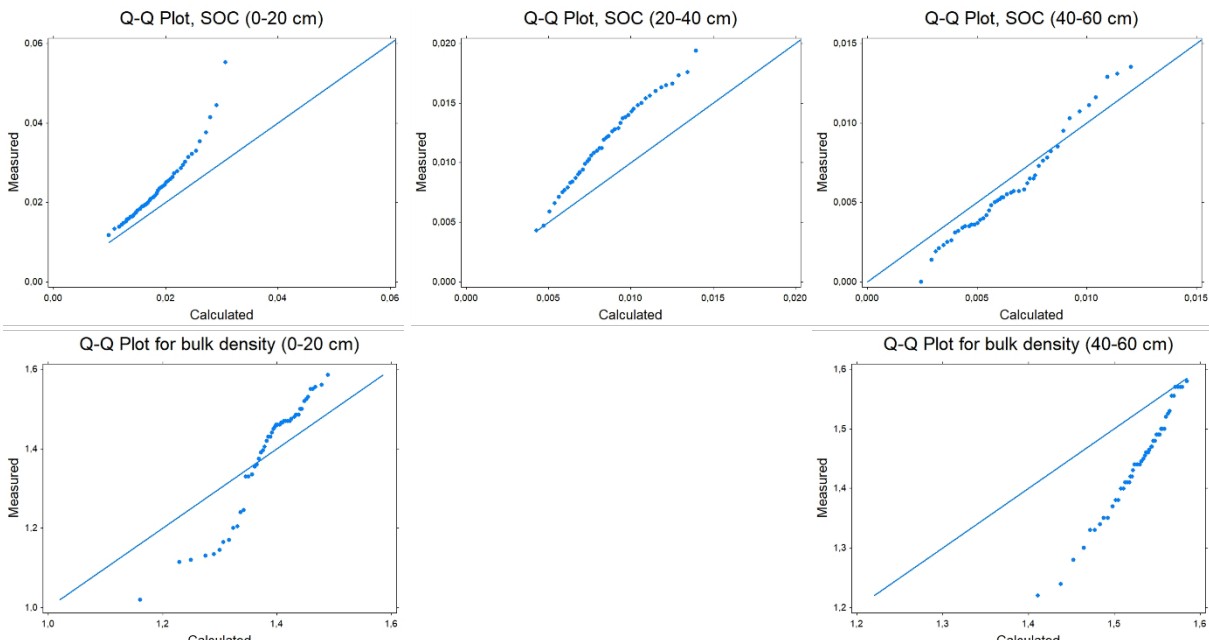

**Figure A1: Q-Q plots for SOC contents (upper row) and soil bulk density (lower row) measured at three and two depths respectively, for soil profiles located in east-central Sweden (production area number 4; Eriksson et al., 2010) with calculated values obtained in the model sensitivity analysis. Note the differences in the y-axis scales for soil organic carbon contents.**

**Code availability**

The model, which is built in the modelling software tool STELLA and the EXCEL data file developed for the sensitivity analysis can be made available on request.

**Data availability**

The data used in the model application to the Ultuna Long-term Organic Matter trial can be obtained by following the link to an on-line repository given in Pold et al. (2025).

**Interactive computing environment -**

**Sample availability -**

**Video supplement -**

**Supplement link -**

**Team list -**

**Author contribution**

**E. COUCHENEY**: Conceptualization; methodology; writing – original draft; writing - review and editing. **A. HERRMANN**: Investigation; data curation; funding acquisition; writing - review and editing. **N. JARVIS**: Conceptualization; formal analysis; funding acquisition; methodology; project administration; writing – original draft; writing - review and editing.

**Competing interests**

The contact author has declared that none of the authors has any competing interests.

**Disclaimer -**

**Special issue statement**

Advances in dynamic soil modelling across scales

**Acknowledgements**

The work described in this paper was funded by the EU EJP SOIL project MaxRootC ("Optimizing roots for sustainable crop production in Europe – pure cultures and cover crops") under the H2020 Grant agreement number 862695 and by the Swedish Research Council for Sustainable Development (FORMAS, grant 2022-00214). Maintenance of the Ultuna Long-Term Field Experiment (RAM-56) is funded by the Faculty of Natural Resources and Agricultural Sciences at the Swedish University of Agricultural Sciences (SLU, Sweden). We would also like

to thank Lorenzo Menichetti (formerly at SLU Department of Ecology, now at LUKE, The Natural Resources Institute of Finland) and Frédéric Rees (INRAE, France) for helpful discussions on this work.

**Financial support**

1-Name of funder: Horizon 2020 & Grant agreement or award number: Grant agreement number 862695

2-Name of funder: Svenska Forskningsrådet Formas & Grant agreement or award number: grant 2022-00214

**Review statement -**

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
