# Peer review of "A simple model of the turnover of organic carbon in a soil profile: model test, parameter identification and sensitivity"

_EGUsphere, 2024_

## Referee Comment (RC1)

The study introduces a parsimonious soil organic carbon (SOC) turnover model for the soil profile that includes key processes controlling carbon persistence. The model specifically incorporates two crucial mechanisms often omitted in simpler models: (1) microbial energy limitation (a form of positive/negative priming where decomposition slows if substrate is scarce) and (2) physical protection via soil aggregation (which protects organic matter from decomposition). The aim was to test this model against long-term field data, identify how well model parameters can be determined (parameter identifiability), and analyze sensitivity of the model to its parameters. However, most of the parameters were not constrained and covariance between the parameters were used as an explanation to parameter unidentifiable. The manuscript is well-written and please find the specific comments as follows:

Line 34, 36: What soil depths represent topsoil and subsoil? Providing the depth ranges would be helpful for clarity.

Line 45-49: The study demonstrated parameter uncertainty and equifinality, despite having fewer parameters than complex models. However, this raises the question: how is the simple model used in this study different from detailed mechanistic models if parameters remain unconstrained? An explanation would help readers understand the trade-offs between complexity and parameter uncertainty.

Line 70: What does ICBM stand for? Additionally, the reference to Andrén and Kätterer (1997) is missing. A schematic diagram illustrating model development over time would help readers visualize how the model has evolved. Similarly, a conceptual diagram of the final model used in this study would be beneficial alongside the mathematical equations.

Line 233: Why were parameters in Table 1 fixed, while only parameters in Table 2 were used in the calibration?

Line 253-254: Why was the mean model efficiency (EF) across all three treatments used to identify acceptable parameter sets?

- Does this mean that the same parameter set was used for all treatments after calibration?

- Why not use treatment-specific parameter sets?

- Wouldn't taking the mean EF lose treatment-specific information that could be valuable for refining the model?

Line 257: 15 model parameters (Table 3?)

Line 320, 329,332, 346, 352: Graphs texts are too small and difficult to read.

Line 314: Given that most parameters were not well constrained, could parameter covariance be a model artifact or a coincidence?

- Could the current dataset be insufficient to constrain these parameters?

- Would incorporating additional data sources (e.g., isotope data, incubation experiments) help resolve this issue? If yes, how can this modeling work be robust?

Line 316: Why were only the 30 best parameter sets selected?

- What was the acceptance rate of parameter sets out of 12,000 simulations?

Line 317: The phrase "strong correlation" is used, but no statistical analysis (e.g., correlation coefficients, p-values) is provided to support this claim. Including quantitative analysis would strengthen this statement.

Line 338: Figures 5 and 6 are difficult to interpret.

- A more detailed explanation of what these figures represent would improve clarity.

- What key insights should the reader take from these figures?

Line 345: Minor inconsistency: (e.g., consistently use "Figure X" or "Fig. X rather than mixing "Fig. X" and "Figure X").

---

## Author Comment (AC1)

**Response to comments from Reviewer 1**

**Authors:**

We would like to thank the reviewer for the positive and perceptive comments, as well as for the questions, which will help us to improve the paper, specifically by giving more details, information and explanations about our data, methods and results.

Reviewer:

The study introduces a parsimonious soil organic carbon (SOC) turnover model for the soil profile that includes key processes controlling carbon persistence. The model specifically incorporates two crucial mechanisms often omitted in simpler models: (1) microbial energy limitation (a form of positive/negative priming where decomposition slows if substrate is scarce) and (2) physical protection via soil aggregation (which protects organic matter from decomposition). The aim was to test this model against long-term field data, identify how well model parameters can be determined (parameter identifiability), and analyze sensitivity of the model to its parameters. However, most of the parameters were not constrained and covariance between the parameters were used as an explanation to parameter unidentifiable.

The manuscript is well-written and please find the specific comments as follows:

**Comment 1**

Line 34, 36: What soil depths represent topsoil and subsoil? Providing the depth ranges would be helpful for clarity.

**Authors:** Here, we define the topsoil as the cultivated (tilled) soil layer, rather than in terms of depth ranges.

We will clarify this in the revised version by writing: "*Most model applications to date have focused on the **cultivated** topsoil, which is clearly of major importance with respect to the effects of soil management on SOC and soil health*"

This will also be added at line 54 "*Furthermore, these models have almost exclusively been tested using measurements in **cultivated** topsoil*"

**Comment 2**

Line 45-49: The study demonstrated parameter uncertainty and equifinality, despite having fewer parameters than complex models. However, this raises the question: how is the simple model used in this study different from detailed mechanistic models if parameters remain unconstrained? An explanation would help readers understand the trade-offs between complexity and parameter uncertainty.

**Authors:** Yes, this is a good point. It's true that even the simplest models of organic matter turnover in soil can show equifinality, depending on the type, quantity and quality of the data used to constrain them. This has been demonstrated for models that are even simpler than the one we developed and tested in our study (see e.g. Juston et al., 2010; Luo et al., 2017). We were therefore expecting to encounter the issue, as we wrote at line 233. This is also why we wrote "*may*" on line 48.

However, in contrast to more complex models, we can clearly see why and where the equifinality arises in our relatively simple model: it depends on the model structure, with correlations among only a few parameters, which makes the problem of parameter uncertainty more manageable. This was the case for the model application to the data at the Ultuna long-term soil organic matter experiment presented in the paper: here, simultaneous calibration to data was sufficient to effectively constrain the model parameters in three treatments with strongly contrasting inputs of

OM with respect to both type and amount. This is what we concluded at lines 317-318: *"These strong correlations of $k_O$, $A_a$ and $k_{till}$ with $\varepsilon$ mean that, in practice, all four parameters are well constrained by the calibration".*

In the revised version, we will clarify these issues by revising and adding to the text at lines 50 to 55:

*" …. their reliability in extrapolation (i.e. prediction of independent data) has not yet been convincingly demonstrated (Garsia et al., 2023; Le Noë et al., 2023). This is because these models have often been tested against insufficient datasets (i.e. observations of topsoil C dynamics at a single site and treatment) which increases the likelihood of equifinality despite the small number of parameters (e.g. Juston et al., 2010; Luo et al., 2017). This may be overcome by simultaneous calibration of the model against data for two or more contrasting treatments, for example with respect to the type and quantity of organic matter inputs (e.g. Meurer et al., 2020) or by multi-site calibration at larger scales using data from long-term field trials at locations with contrasting soils and management practices (e.g. Juston et al., 2010; Dechow et al., 2019). Testing model predictions for entire soil profiles remains however difficult and is therefore rarely done, because fewer measurements are made in subsoils and the turnover of organic C in subsoil is very slow, so datasets will rarely be long enough to detect any changes. Additional data sources may also help to alleviate problems arising from equifinality. One possibility is to make use of … "*

**Comment 3**

Line 70: What does ICBM stand for? Additionally, the reference to Andrén and Kätterer (1997) is missing. A schematic diagram illustrating model development over time would help readers visualize how the model has evolved. Similarly, a conceptual diagram of the final model used in this study would be beneficial alongside the mathematical equations.

**Authors:** ICBM stands for *Introductory Carbon Balance Model*. We apologize for not including the reference to Andrén and Kätterer (1997). We will add this to the revised version in line 70.

We will also include a conceptual diagram of the model to go alongside the equations.

**Comment 4**

Line 233: Why were parameters in Table 1 fixed, while only parameters in Table 2 were used in the calibration?

**Authors:** Some reasons were already given at lines 234 to 239. However, we realize now that this description was insufficient. We will add some new text as well as supporting references in tables 1 and 2 in the revised version to mention these additional reasons. At line 231, the text will be revised to:

*"The model was simultaneously calibrated to the measurements from the three treatments using the Generalized Likelihood Uncertainty Estimation (GLUE) method (Beven, 2006; Beven and Binley 2014; Juston et al., 2010) …. Six of the fifteen parameters were included in this analysis (Table 1) as they were judged to be uncertain and sensitive. The remaining nine parameters were set to fixed values (Table 2) as they could be estimated from measurements (e.g. $f_{clay}$, $f_{agg}$, $F_p$) or they were not expected to be sensitive (e.g. $k_y$, Andrén and Kätterer, 1997; Juston et al., 2010; Meurer et al., 2020), or both (e.g. $\psi_{ae}$, $\psi_{mic}$, $\phi_{min}$, $\gamma_o$, $\gamma_m$)".*

**Comment 5**

Line 253-254: Why was the mean model efficiency (EF) across all three treatments used to identify acceptable parameter sets?

**Authors:** Because we wanted to obtain a common parametrization for all three treatments. We will clarify this in the revised version. We will add text at L254: "*This was done in order to obtain a more robust parameterization of the model by selecting only parameter sets that simultaneously fitted all three treatments well.*"

- Does this mean that the same parameter set was used for all treatments after calibration?

  **Authors:** Yes. This was implied earlier at lines 231 to 232. However, in the revised version, we will clarify this by writing explicitly that we wanted to obtain a common parametrization for all three treatments.

- Why not use treatment-specific parameter sets?

  **Authors:** This is as stated above to obtain a robust model parameterization of the model, i.e. parameters sets that should be valid for a wide range of conditions. This is also a way to reduce issues with parameter equifinality.

  Also, if parameter values must be changed to account for different treatments (in this case, the amount and type of organic matter inputs), it is a sure sign that something important is missing or wrong in the model, with respect to process descriptions. This means, in turn, that predictions made for contrasting conditions (re. OM inputs) using those calibrated parameter sets may be wildly wrong.

  We wanted to critically test the model to see whether it could match the data from the three treatments with a common parameterization. It passed this test.

- Wouldn't taking the mean EF lose treatment-specific information that could be valuable for refining the model?

  **Authors:** In principle, yes, it would, but only if the model had performed poorly. However, as shown in figure 2, and as we wrote at lines 308 to 310 (and in the abstract at lines 15 to 20), we were able to get excellent calibrated fits to the data from the three treatments with exactly the same parameterization. This demonstration of the capability of such a simple model is an important result of this study.

**Comment 6**

Line 257: 15 model parameters (Table 3?)

**Authors:** Yes, the 15 parameters in table 3 were included in the sensitivity analysis. We will refer to Table 3 in the text.

Line 320, 329,332, 346, 352: Graphs texts are too small and difficult to read.

**Authors:** OK, yes, we will make revised versions of these figures with more legible text

**Comment 7**

Line 314: Given that most parameters were not well constrained, could parameter covariance be a model artifact or a coincidence?

**Authors:** It is a consequence of the model structure, so not a coincidence. We feel that this is already stated quite clearly at lines 314 to 316.

(note that we would not like to call this a model *artifact*, because it is inherent to the model and not something that occurs by chance as a consequence of the calibration methods applied)

- Could the current dataset be insufficient to constrain these parameters?

  **Authors:** This is a good point: to some extent perhaps. For example, if we had the same kind of data from tilled and untilled soils (so 6 treatments at the site, three different OM inputs, with and without tillage), then we would probably have been able to more clearly identify the parameter $k_{till}$ but only under the condition that the model describe the effects of tillage in a reasonable way (which we don't know yet).

- Would incorporating additional data sources (e.g., isotope data, incubation experiments) help resolve this issue? If yes, how can this modeling work be robust?

  **Authors:** We mentioned in the introduction (L55-58) that *in situ* isotope data is also a possible way to reduce some of the model parameter uncertainty, under the condition that they would be prove to be sensitive. We are more doubtful about incubation data based on disturbed/sieved soil samples. We think that the approach adopted here is anyway rather robust, as it included data from several treatments simultaneously.

  It is also worth noting that this is the first application of a new model. We wrote at line 404 in the section "*Concluding remarks*" that the tests of the model in this paper suggest that it "*shows promise*". We don't think this claim is unreasonably strong. And as we wrote at lines 407 to 409, a greater degree of confidence in the robustness of the model can, of course, be established over time by showing that it produces acceptable results when repeatedly tested against different data sets.

**Comment 8**

Line 316: Why were only the 30 best parameter sets selected?

**Authors:** Because their predictions were sufficient to cover the range of variability observed in the measurements which is a criteria for the GLUE method. This was implied in the text at lines 291 to 293, but it was not explained. We will mention this criteria in the M&M section after lines 231-232.

*"The model was simultaneously calibrated to the measurements from the three treatments using the Generalized Likelihood Uncertainty Estimation (GLUE) method (Beven, 2006; Beven and Binley 2014; Juston et al., 2010). The number of acceptable parameter sets was determined such that the range of variation of their predictions approximately covered the variations observed in the measurements. With this criterion, 30 of the 12000 parameter sets were identified as acceptable".*

- What was the acceptance rate of parameter sets out of 12,000 simulations?

  **Authors:** 30 out of 12000 = 0.25%. This small value is a consequence of the inefficient sampling which is inherent in the GLUE method: note that it says nothing about the quality of the model. We can state this in a follow-up sentence:

  *"Note that this low acceptance rate is a consequence of the inefficient sampling inherent to the GLUE method and says nothing about the quality of the model."*

**Comment 9**

Line 317: The phrase "strong correlation" is used, but no statistical analysis (e.g., correlation coefficients, p-values) is provided to support this claim. Including quantitative analysis would strengthen this statement.

**Authors:** Yes, we will add $R^2$ and p-values in a revised version of the figure. All four relationships are highly significant (p<0.002) with $R^2$ values varying between 0.30 and 0.76.

**Comment 10**

Line 338: Figures 5 and 6 are difficult to interpret.

- A more detailed explanation of what these figures represent would improve clarity.

- What key insights should the reader take from these figures?

**Authors:** Yes, we agree that this was not well explained. In addition to what we wrote at lines 336 to 338, these figures also suggest that the results of the sensitivity analysis should be *"reasonably well grounded in reality"*. We wrote this at line 343 in connection with figure 7, but this conclusion should also be based on figures 5 and 6. In the revised version, we will modify the text at lines 342 to 343 to make this clearer.

**Comment 11**

Line 345: Minor inconsistency: (e.g., consistently use "Figure X" or "Fig. X rather than mixing "Fig. X" and "Figure X").

**Authors:** Yes, we will fix this in the revised version

**References cited (in answers to both reviewer 1 and 2)**

Andrén, O., Kätterer, T. 1997. ICBM: the introductory carbon balance model for exploration of soil carbon balances. Ecological Applications, 7, 1226-1236.

Chen, S., Zou, J., Hu, Z., Lu, Y. 2020. Temporal and spatial variations in the mean residence time of soil organic carbon and their relationship with climatic, soil and vegetation drivers. Global and Planetary Change, 195, 103359.

Coucheney, E., Kätterer, T., Meurer, K.H.E., Jarvis, N. 2024. Improving the sustainability of arable cropping systems by modifying root traits: a modelling study for winter wheat. European Journal of Soil Science, 75, e13524.

Dechow, R., Franko, U., Kätterer, T., Kolbe, H. 2019. Evaluation of the RothC model as a prognostic tool for the prediction of SOC trends in response to management practices on arable land. Geoderma, 337, 463-478.

Kätterer, T., Bolinder, M. 2024. Response of maize yield to changes in soil organic matter in a Swedish long-term experiment. European Journal of Soil Science, 75, e13482.

Juston, J., Andrén, O., Kätterer, T., Jansson, P-E. 2010. Uncertainty analyses for calibrating a soil carbon balance model to agricultural field trial data in Sweden and Kenya. Ecological Modelling, 221, 1880-1888.

Luo, Z., Wang, E., Sun, O. 2017. Uncertain future soil carbon dynamics under global change predicted by models constrained by total carbon measurements. Ecological Applications, 27, 1001-1009.

Meurer, K., Chenu, C., Coucheney, E., Herrmann, A., Keller, T., Kätterer, T., Nimblad Svensson, D., Jarvis, N. 2020. Modelling dynamic interactions between soil structure and the storage and turnover of soil organic matter. Biogeosciences, 17, 5025-5042

Poeplau, C., Don, A., Schneider, F. 2021. Roots are key to increasing the mean residence time of organic carbon entering temperate agricultural soils. Global Change Biology, 27, 4921–4934.

Pold, G., MacDonald, E., Braun S., Herrmann, A. M., 2025. Soil and vegetation property data from the Ultuna R3-RAM56 long-term soil amendment experiment, 1956-2023

Wutzler, T., Reichstein, M. 2013. Priming and substrate quality interactions in soil organic matter models. Biogeosciences, 10, 2089-2103.

---

## Author Comment (AC2)

**Response to comments from Reviewer 2**

**Authors:**

We would like to thank the reviewer for the positive and perceptive comments, as well as for the questions, which will help us to improve the paper, specifically by giving more details, information and explanations about our data, methods and results.

This study describes a simple model of soil organic carbon (SOC) turnover that represents the effects of soil physical protection and microbial energy limitation. The paper first describes the model in a soil profile, then tests it using SOC data from a long-term study on agricultural fields with varied C inputs to the soils. Finally, the model's most influential parameters were identified in a sensitivity and uncertainty analysis. Overall, the paper is well written and presents a model of interest and relevance to soil carbon management. Below, please find specific comments intended to help improve the paper.

**Comment 1**

L83 – Define the abbreviation USSF.

**Authors:** OK, we will do this in the revised version (it stands for **U**ppsala model of **S**oil **S**tructure and **F**unction)

**Comment 2**

L91-95 – I would suggest ending the introduction with a strong thesis statement of what the paper contributes to current knowledge of the subject.

**Authors:** Yes, we will add such a statement at the end of the introduction.

*"The contribution of this study is to demonstrate the utility of a simple soil C turnover model that can account for the nexus of soil management, soil structure and microbial activity that critically determine C mineralization and stabilization at the scale of a soil profile."*

**Comment 3**

L97-101 – I appreciate this overview of the methods, very helpful to have this framework.

**Authors:** Thanks!

**Comment 4**

L103 – In section 2.1.1 that starts on this line, it is unclear if the model as described in this section is the work of the authors or if this is describing previously published work. If it has been previously published, I suggest including most of the equations in this section in a supplement rather than in the main document. In the main manuscript, I suggest describing the model in writing and including important equations for the modifications to the model that are new in the current study.

**Authors:** The model is based on a combination of the model described by Meurer et al. (2020) accounting for physical protection in relation to soil properties with the model described in Wutzler and Reichstein (2013) for microbial energy limitation. This combination of the two models was outlined in Coucheney et al. (2024), along with some minor improvements and modifications. However, the description of this SOM model was only included in the supplementary information in Coucheney et al. (2024) as the model itself was not tested at all. So although the main constituent components of the model have been described earlier, this is the first time that the complete model has been *tested*. As the model is quite new and previously untested, we prefer to keep these equations in the main paper, as it makes it easier for the reader. The paper is not too long and including the equations in the main text will ensure that the equations are readily available.

We think this history of the model development is clearly explained at lines 104 to 115.

Additionally, I would encourage the authors to post their full model code online and cite it in the paper.

**Authors:** We built the model using the icon-based modelling software STELLA, which is a commercial product. The model file will be made available on request to the authors – this will be stated in the "data availability statement" at the end of the manuscript.

**Comment 5**

L107-109 – It would be helpful to specify the direction of the relationship between these effects (e.g., Do smaller pores get fewer root derived inputs?

**Authors:** Not necessarily, no. It depends on the pore size distribution in the soil, which in turn depends on the soil texture (or more simply clay content in our approach). The pore size distribution determines the partitioning of root-derived inputs of OM between the two pore regions (see Eqs. 10-13). This means that a larger proportion of the root C inputs would enter the micropore region in a clay soil than in a sandy soil, because the porosity of a clay soil predominantly consists of smaller pores. Clay soils therefore have a higher potential for physical protection of soil C. The effect of the physical protection is quantified by the factor $F_p$ (Eqs. 1-4) that reduces the rate of SOC decomposition in micropores.

We will modify the text at L107-109 to make this clearer:

*"In turn, the pore size distribution determines the partitioning of root-derived inputs of OM between the two pore regions. Compared with a sandy soil, a larger proportion of the root OM input will enter the micropore region in a clay soil, as it predominantly consists of smaller pores. The soil pore size distribution also regulates decomposition rates with slower decomposition rates of OM stored in microporous regions of the soil. Compared with sandy soils, clay soils therefore have a greater potential for physical protection of soil C".*

Do micropores have lower decomposition rates?).

**Authors:** Yes, they do. We will clarify this in the revised text at lines 107-109 (see above text)

**Comment 6**

L125 – Is the "(-)" after fr,mic supposed to indicate that it is unitless?

**Authors:** Yes

**Comment 7**

L218 – How does this straw addition rate compare to the maize biomass per hectare?

**Authors:** Note that maize was only grown since 2000. Between 2000-2019, the crop fertilized and crop fertilized+straw treatments had on average 6.07 and 7.09 t ha$^{-1}$ maize yield, respectively (see Kätterer & Bolinder, 2024). However, above-ground maize and crop residues were removed from the field and it is very speculative how much C is added to the soil system via root inputs (rhizodeposition and exudates) and such C inputs are challenging to quantify and are therefore not included in the present proposed model. For the modelling, we know how much straw was added and this is included in the paper.

**Comment 8**

L219 - 220 – Include the scientific names/varieties of the crops.

**Authors:** We will include Latin names in the revised version:

"Maize (*Zea mays*) has been grown on the cropped plots since 2000. Before 2000, the crop rotation included barley (*Hordeum vulgare*), oats (*Avena sativa*), beets (*Beta vulgaris*) (prior to 1967) and rape (*Brassica napus*)."

In addition, we propose to include a reference to a newly published data paper which gives a complete description of the experiment, as well as links to all the data (Data in Brief paper. Pold et al. 2025 - Soil and vegetation property data from the Ultuna R3-RAM56 long-term soil amendment experiment, 1956-2023).

**Comment 9**

L220 - What is the purpose of hand digging the plots after harvest?

**Authors:** It's to simulate ploughing: the plot size of 4 m² is too small to manage in the same way as a farmer's field. We will add this information at L220.

**Comment 10**

L 231 – What measurements were used for the calibration? Only OM or additional measurements?

**Authors:** Only SOC. We will state this explicitly in the revised version.

Calibrating to additional variables could help reduce equifinality.

**Authors:** Perhaps, depending on the type of data and its quantity and quality. But we were able to strongly reduce the prior uncertainty ranges for the parameter values anyway, despite parameter correlation.

See also our answer to comment number 7 from reviewer 1.

**Comment 11**

L238 – Specify the field bulk density values used for this validation.

**Authors:** OK, yes, we will do so (the numbers are 1.43, 1.28 and 1.21 g/cm3 in the treatments "Fallow", "N-fertilized" and "N fertilized+straw" respectively)

**Comment 12**

Table 1 – Are there field measurements available for any of these parameters? If so, how similar are they to these fixed values?

**Authors:** Yes, we didn't write it explicitly but these values in table 1 were based (with one exception) on data obtained at the study site. We will add a column to this table with the heading "Source" where we will cite the relevant studies.

See also answer to comment 4 – reviewer 1

**Comment 13**

Figure 1 needs a legend to identify what the different colors/patterns of shading indicates.

**Authors:** We will add this information to the caption

*"Figure 1. Map of Sweden **(in white)** showing the location of the Ultuna Long-term Soil Organic Matter Experiment (Uppsala, Sweden) and the extent of the production area PO4 **(shaded area in grey)**. Drawn by Anna Lindahl, SLU from Esri, TomTom, Garmin, FAO, NOAA, USGS"*

**Comment 14**

Table 1 – Are there data references for these parameter values? If not, how did you come to these values?

**Authors:** Yes, we will give these references. We will add a column to this table with the heading "Source"

Tables – I recommend including captions for tables with relevant details.

**Authors:** We will revise the table captions giving more informative detail. For example:

*"Table 1. List of the 9 model parameters that were fixed at constant values during the calibration based on field measurements or literature data"*

*"Table 2. List of the 6 selected model parameters included in the model calibration to the Ultuna Long-Term Soil Organic Matter Experiment and their initial parameter uncertainty ranges"*

*"Table 3. List of the 15 model parameters and the input distributions used in the sensitivity analysis"*

**Comment 15**

L264- What determined if the data support was sufficient?

**Authors:** This is partially a subjective decision. However, we can revise the sentence to be a bit more specific:

*"We assumed normal distributions when the **literature or measured** data was considered sufficient **to support such a distribution**, while uniform distributions were used otherwise (Table 3). "*

**Comment 16**

Table 3 – The source "SCB Statistics Sweden" needs to be more specific. Same comment for source listed as "site data" - where are site data accessible? Year, dataset name, authors etc.

**Authors:** Yes, we will add these details

**Comment 17**

L304 - 306 – Is this distinction of straw going into only mesopores and root OM going partially into micropores supported by empirical data?

**Authors:** No, it's more a model hypothesis, although based on process understanding. We can't see how digging or ploughing down above-ground crop residues could possibly incorporate these residues into the microporous regions of the soil where pore diameters are less than 5 microns (in this study). In contrast, roots can grow through microporous soil regions supplying both POM (on root death) and root exudates.

This hypothesis could be tested experimentally in future work with for example, the help of X-ray tomography on samples taken soon after harvest and ploughing-down of above-ground crop residues). This is however way beyond the scope of the present study.

**Comment 18**

L308 – What is meant by "export of residues?" Does that mean the removal of residues by land managers?

**Authors:** Yes. In this treatment, most of the above-ground crop residues are removed. This was explained at line 217. We will replace this phrase by … "**near total** removal of above-ground crop residues from the field"

**Comment 19**

Figure 3 – This panel figure needs letters for each panel and a description of each panel and the definition of the X axes in the figure caption.

**Authors:** We feel that letters for the panels are not needed but we will give further details in the figure caption instead:

*"Figure 3. Mean model efficiencies for each simulation (i.e. parameter set) in the GLUE analysis that gave values larger than zero, plotted against the values of the six parameters included in the analysis (refer to Table 2 for parameter definitions and descriptions)."*

Figure 4 – This panel figure also needs to have the individual panels labeled/described and the axes defined in the caption.

**Authors:** We feel that letters for the panels are not needed because we don't refer to individual subfigures in the text, but we will give further details in the figure caption instead:

*"Figure 4. Inter-relationships among four (out of six) of the targeted model parameters in the calibration. Relationships are shown for the 30 best parameter sets identified in the GLUE analysis (refer to Table 2 for parameter definitions and descriptions)".*

**Comment 20**

L314 – In the introduction, large mechanistic models are criticized for having uncertainty and equifinality. That was provided as justification for a simpler parsimonious model. How does that criticism relate to your finding that the simple/parsimonious model presented here has the same issue of equifinality and parameter uncertainty as the more complex models? How does this affect the usefulness of the model or its applicability compared to the larger models? Or, should this model's parameters be further simplified?

**Authors:** Please see our answer to comment 2 of reviewer 1 (also copied here)

Yes, this is a good point. It's true that even the simplest models of organic matter turnover in soil can show equifinality, depending on the type, quantity and quality of the data used to constrain them. This has been demonstrated for models that are even simpler than the one we developed and tested in our study (see e.g. Juston et al., 2010; Luo et al., 2017). We were therefore expecting to encounter the issue, as we wrote at line 233. This is also why we wrote "*may*" on line 48.

However, in contrast to more complex models, we can clearly see why and where the equifinality arises in our relatively simple model: it depends on the model structure, with correlations among only a few parameters, which makes the problem of parameter uncertainty more manageable. This was the case for the model application to the data at the Ultuna long-term soil organic matter experiment presented in the paper: here, simultaneous calibration to data was sufficient to effectively constrain the model parameters in three treatments with strongly contrasting inputs of OM with respect to both type and amount. This is what we concluded at lines 317-318: *"These strong correlations of $k_O$, $A_a$ and $k_{till}$ with $\varepsilon$ mean that, in practice, all four parameters are well constrained by the calibration".*

In the revised version, we will clarify these issues by revising and adding to the text at lines 50 to 55:

*" …. their reliability in extrapolation (i.e. prediction of independent data) has not yet been convincingly demonstrated (Garsia et al., 2023; Le Noë et al., 2023). This is because these models have often been tested against insufficient datasets (i.e. observations of topsoil C dynamics at a single site and treatment) which increases the likelihood of equifinality despite the small number of parameters (e.g. Juston et al., 2010; Luo et al., 2017). This may be overcome by simultaneous calibration of the model against data for two or more contrasting treatments, for example with respect to the type and quantity of organic matter inputs (e.g. Meurer et al., 2020) or by multi-site calibration at larger scales using data from long-term field trials at locations with contrasting soils and management practices (e.g. Juston et al., 2010; Dechow et al., 2019). Testing model predictions for entire soil profiles remains however difficult and is therefore rarely done, because fewer measurements are made in subsoils and the turnover of organic C in subsoil is very slow, so datasets will rarely be long enough to detect any changes. Additional data sources may also help to alleviate problems arising from equifinality. One possibility is to make use of … "*

**Comment 21**

L317 – What is the evidence of strong correlation? This statement needs statistical support.

**Authors:** Yes, we will add $R^2$ and p-values in a revised version of figure 4. All four relationships are highly significant (p<0.002) with $R^2$ values varying between 0.30 and 0.76.

**Comment 22**

L341 – Temperate, not temperature.

**Authors:** Thanks, we will fix this

**Comment 23**

Figure 5 – Can you provide a quantitative comparison of the means to support the conclusions?

**Authors:** This is a good point. We didn't attempt any statistics, simply writing that the model "*gives reasonably realistic predictions*". In fact, statistical tests for differences between the two distributions (not just the means) show that they are significantly different, which is almost entirely due to the much larger spread in the measurements compared to the calculations, especially the occurrence of a number of outliers with large values of organic carbon contents and correspondingly small values of bulk density. This is not really surprising because the calculations do not include the effects of all factors affecting SOC and bulk density. Our guess is that the large values of organic carbon content (and small values of bulk density) correspond to locations with wet soils due to topography (i.e. flood plains, depressions). The model, as it is formulated here, does not include the effects of excess soil moisture on decomposition rates.

We will add some new text in the revised version at line 338 to explain the above.

*"A qualitative comparison with soil survey data for agricultural land in east-central Sweden (production area PO4) suggests that despite its simplicity the model estimates of steady-state SOC and bulk density in the soil profile lie mostly within the range of variation encountered in the region (Fig. 5 and Fig.6). Nevertheless, statistical tests (Z-test) show that the distributions of simulated and measured values of SOC and bulk density are different, which is almost entirely due to the much larger spread in the measurements compared to the calculations and especially the occurrence of a number of outliers with large values of organic carbon contents and correspondingly small values of bulk density. This is not really surprising because the calculations do not include the effects of all factors affecting SOC and bulk density. The large values of organic carbon content (and small values of bulk density) almost certainly correspond to locations with wet soils due to topography (i.e. flood plains, depressions). The model, as it is formulated here, does not include the effects of excess soil moisture on decomposition rates."*

We will avoid using the word "*predictions*", because these are the aggregated outputs of a sensitivity analysis and not model predictions that can be compared with measurements at specific locations.

**Comment 24**

Figure 6 – Why are only two depths shown in this figure? (there are 3 depths in the previous figure)

**Authors:** Because we only have bulk density data at two depths, i.e. between 0-20 cm depth and 40-60 cm depth, whereas SOC was measured between 0-20, 20-40 and 40-60 cm depth (see line 281)

L357 – What are the cutoffs for NRC values to determine if they are strongly, moderately, or minimally sensitive in this analysis? I think that information should be included in the methods. In the discussion, it would be helpful to more quantitatively compare/describe the model sensitivity to these various parameters.

**Authors:** We prefer to discuss these results in relative terms and therefore we choose to present all values ordered decreasingly in Table 5 without any use of defining cut off values that may be judged to be arbitrary (the readers can also see all of the figures and apply their own criteria if wanted)

**Comment 25**

L359 – The fraction of aboveground residues incorporated is roughly as important as the clay content yet it is not mentioned here.

**Authors:** Yes, we agree that it should also be mentioned. We will include a mention of $f_{inc}$ at lines 361 to 363, alongside the other two parameters that determine the input of above-ground residues.

I'm also unclear why the clay content is mentioned before the other more sensitive parameters in the table.

**Authors:** Yes, we see your point. We mentioned the clay content here because it belongs together with $F_p$ in that taken together they determine the extent of physical protection. We did try to explain this at line 360, but we will make this connection much clearer in the revised version:

*"The soil clay content, which **together with $F_p$, determines** the extent to which physical protection is expressed in soils of contrasting texture, is also a relatively sensitive model parameter, **albeit to a lesser extent (Table 5)."***

**Comment 26**

L367-370 – I would be interested to see this idea expanded upon – how does this USSF model result relate back to your model result of an 8% increase in SOM?

**Authors:** It's really mostly just a consequence of the fact that it takes a lot longer than 30 years to reach steady-state after a change in OM inputs. We can add some text to explain this.

And what are the larger implications of these increases for climate change mitigation as you mention? For example, how does a 1.4% increase over the course of 30 years compare to targeted goals for mitigation?

**Authors:** This is discussed in Coucheney et al. (2024) and we refer to this study in the text. We don't think this is the right place to discuss the results from a previous paper.

**Comment 27**

L379 – Here, the authors seem to consider a 4-5% reduction in SOM to be minor. But on lines 374 - 377 they seem to indicate that a 3-5% increase is significant. Some benchmarks for the relevance of these changes would be helpful for interpretation of the results.

**Authors:** Thanks for pointing this out. It is only an apparent inconsistency, because the 3-5% increase mentioned at lines 374-377 was after 30 years, whereas at line 379 we are referring to steady-state values. We will make this clearer by slightly modifying the sentence at L377-378:

*"Table 5 suggests that tillage is one of the least sensitive factors affecting **steady-state** SOM stocks "*

**Comment 28**

L381 – Haddaway et al. found a difference by tillage intensity in the topsoil, as opposed to what is stated here. Intermediate intensity tillage resulted in greater SOC stocks than the high intensity tillage in that metanalysis.

**Authors:** This seems to be a misunderstanding. We did write that Haddaway et al. found greater stocks of C in the topsoil under no-till. However, we also wrote that they did not find any difference in total C stocks in the profile between no-till and intensive or intermediate tillage systems, which is also true. We will slightly revise the sentence:

*"…. Haddaway et al. (2017) and Meurer et al. (2018) found larger SOC stocks under no-till compared with conventional tillage in the topsoil, but no overall difference in total SOC stocks in the soil profile down to 60 cm depth."*

**Comment 29**

L385 – It would be helpful to include empirical data supporting these model data on the figure for comparison with the model data. Or include the empirical data in a table caption.

**Authors:** We would prefer not to modify the actual figure to show data from other studies. This is partly because a direct comparison of predictions for a region in east-central Sweden with measurements from a highly diverse global dataset like Chen et al. (2020) could give readers a misleading impression. In addition, the raw data on MRT from the study by Poeplau et al. (2021) are not available to us. The value quoted in the paper at line 339, "ca. 20 years" was taken from a table in the paper (table 1). In the revised version of the paper, we will also add the values reported by Chen et al. (2020) for cropland to this text, as this data is available. In doing so, we will also slightly modify the text at lines 341 to 342: *"…. and they also lie at the high end of the range reported by Chen et al. (2020) for cropland (mean = 9.5 years, standard deviation = 6 years; n = 217)*. We feel this is better than adding these numbers to the figure caption.

**Comment 30**

Table 5 – It looks like this table was color coded according to the colors of the groups in table 3. That information and the meaning of the colors should be included in the caption.

**Authors:** Yes, that's right. However, we have been advised by the editor that we must remove this colour coding in the revised version of the paper.

**References cited (in answers to both reviewer 1 and 2)**

Andrén, O., Kätterer, T. 1997. ICBM: the introductory carbon balance model for exploration of soil carbon balances. Ecological Applications, 7, 1226-1236.

Chen, S., Zou, J., Hu, Z., Lu, Y. 2020. Temporal and spatial variations in the mean residence time of soil organic carbon and their relationship with climatic, soil and vegetation drivers. Global and Planetary Change, 195, 103359.

Coucheney, E., Kätterer, T., Meurer, K.H.E., Jarvis, N. 2024. Improving the sustainability of arable cropping systems by modifying root traits: a modelling study for winter wheat. European Journal of Soil Science, 75, e13524.

Dechow, R., Franko, U., Kätterer, T., Kolbe, H. 2019. Evaluation of the RothC model as a prognostic tool for the prediction of SOC trends in response to management practices on arable land. Geoderma, 337, 463-478.

Kätterer, T., Bolinder, M. 2024. Response of maize yield to changes in soil organic matter in a Swedish long-term experiment. European Journal of Soil Science, 75, e13482.

Juston, J., Andrén, O., Kätterer, T., Jansson, P-E. 2010. Uncertainty analyses for calibrating a soil carbon balance model to agricultural field trial data in Sweden and Kenya. Ecological Modelling, 221, 1880-1888.

Luo, Z., Wang, E., Sun, O. 2017. Uncertain future soil carbon dynamics under global change predicted by models constrained by total carbon measurements. Ecological Applications, 27, 1001-1009.

Meurer, K., Chenu, C., Coucheney, E., Herrmann, A., Keller, T., Kätterer, T., Nimblad Svensson, D., Jarvis, N. 2020. Modelling dynamic interactions between soil structure and the storage and turnover of soil organic matter. Biogeosciences, 17, 5025-5042

Poeplau, C., Don, A., Schneider, F. 2021. Roots are key to increasing the mean residence time of organic carbon entering temperate agricultural soils. Global Change Biology, 27, 4921–4934.

Pold, G., MacDonald, E., Braun S., Herrmann, A. M., 2025. Soil and vegetation property data from the Ultuna R3-RAM56 long-term soil amendment experiment, 1956-2023

Wutzler, T., Reichstein, M. 2013. Priming and substrate quality interactions in soil organic matter models. Biogeosciences, 10, 2089-2103.

---

## Referee Report (RR1)

Dear Editor, please find below a check that the referees' comments have been incorporated satisfactorily in the final revised version of the article submitted by Coucheney et al, entitled "A simple model of the turnover of organic carbon in a soil profile: model test, parameter identification and sensitivity"

The referees' comments are in black, the authors' responses in blue, my comments in red.
* * *
Dear Editors of the SOIL journal,

We hereby present the revised manuscript entitled "A simple model of the turnover of organic carbon in a soil profile: model test, parameter identification and sensitivity", in which we have implemented revisions according to the suggestions and questions of the two reviewers.

This includes

- The inclusion of additional text in the introduction, results and discussion sections to further clarify or specify our approach (for example, how we deal with the equifinality issue in our modelling).

- An additional figure (Figure 1) showing a schematic of the model developed and applied.

- A review of the quality of all figures (and colour changes where appropriate) and table captions (with more specific text).

- In addition, note that former Figures 6 and 7 have been combined into a single figure (now Figure 6), so that Figure 8 has become Figure 7.

- References have been updated and completed. In addition, the format for publication in SOIL has been updated.

- A data availability and model statement has been added at the end of the manuscript.

Below is a detailed description of the revision in response to the Reviewercomments. Line references are to the tracked-changes document (.PDF)
I hope you find this manuscript suitable for publication in SOIL.

Best                                                                                              regards,
Elsa                    Coucheney,                    for                    the                    co-authors

Response                to                comments                from                Reviewer1

Authors:
We would like to thank the Reviewer for the positive and perceptive comments, as well as for the questions, which will help us to improve the paper, specifically by giving more details, information and explanations about our data, methods and results.

Reviewer:
The study introduces a parsimonious soil organic carbon (SOC) turnover model for the soil profile that includes key processes controlling carbon persistence. The model specifically incorporates two crucial mechanisms often omitted in simpler models: (1) microbial energy limitation (a form of positive/negative priming where decomposition slows if substrate is scarce) and (2) physical protection via soil aggregation (which protects organic matter from decomposition). The aim was to test this model against long-term field data, identify how well model parameters can be determined (parameter identifiability), and analyze sensitivity of the model to its parameters. However, most of the parameters were not constrained and covariance between the parameters were used as an explanation to parameter unidentifiable. The manuscript is well-written and please find the specific comments as follows:

Comment 1

Line 34, 36: What soil depths represent topsoil and subsoil? Providing the depth ranges would be                                  helpful                                  for                                  clarity.
Authors: Here, we define the topsoil as the cultivated (tilled) soil layer, rather than in terms of depth ranges.  We will clarify this in the revised version L33, by writing: "Most model applications to date have focused on the cultivated topsoil, which is clearly of major importance with respect to the effects of soil management on SOC and soil health"

REVIEWER: the notion of depths remains unprecise but at this point of the introduction, the proposed change is sufficient to figure out model applications.

Comment 2

Line 45-49: The study demonstrated parameter uncertainty and equifinality, despite having fewer parameters than complex models. However, this raises the question: how is the simple model used in this study different from detailed mechanistic models if parameters remain unconstrained? An explanation would help readers understand the trade-offs between complexity and parameter uncertainty.

Authors: Yes, this is a good point. It's true that even the simplest models of organic matter turnover in soil can show equifinality, depending on the type, quantity and quality of the data used to constrain them. This has been demonstrated for models that are even simpler than the one we developed and tested in our study (see e.g. Juston et al., 2010; Luo et al., 2017). We were therefore expecting to encounter the issue, as we wrote at line 233. This is also why we wrote "may" on line 48.

However, in contrast to more complex models, we can clearly see why and where the equifinality arises in our relatively simple model: it depends on the model structure, with correlations among only a few parameters, which makes the problem of parameter uncertainty more manageable. This was the case for the model application to the data at the Ultuna long-term soil organic matter experiment presented in the paper: here, simultaneous calibration to data was sufficient to effectively constrain the model parameters in three treatments with strongly contrasting inputs of OM with respect to both type and amount. This is what we concluded at lines 317-318: "These strong correlations of ko, Aa and ktill with epsilon mean that, in practice, all four parameters are well constrained by the calibration".

In the revised version, we will clarify these issues by revising and adding to the text at lines 52 to 61: " This is because these models have often been tested against limited datasets (i.e. observations of topsoil C dynamics at a single site and treatment) which increases the likelihood of equifinality despite the small number of parameters (e.g. Juston et al., 2010; Luo et al., 2017). This may be overcome by simultaneous calibration of the model against data for two or more contrasting treatments, for example with respect to the type and quantity of organic matter inputs (e.g. Meurer et al., 2020) or by multi-site calibration at larger scales using data from long-term field trials at locations with contrasting soils and management practices (e.g. Juston et al., 2010; Dechow et al., 2019). Testing model predictions for entire soil profiles remains however difficult and is therefore rarely done, because fewer measurements are made in subsoils and the turnover of organic C in subsoil is very slow, so datasets will rarely be long enough to detect any changes. Additional data sources may also help to alleviate problems arising from equifinality. … "

Reviewer: Generally, the manuscript is now clear concerning equifinality and analysis of parameters obtained via the GLUE procedure, constraints, and correlations.

Comment 3

Line 70: What does ICBM stand for? Additionally, the reference to Andrén and Kätterer (1997) is missing. A schematic diagram illustrating model development over time would help readers visualize how the model has evolved. Similarly, a conceptual diagram of the final model used in this study would be beneficial alongside the mathematical equations. Authors: ICBM stands for Introductory Carbon Balance Model. We apologize for not including the reference to Andrén and Kätterer (1997). We will add this to the revised version in line 79. We will also include a conceptual diagram of the model to go alongside the equations which is referred to Figure 1, page 5.

Reviewer: Please explain what do M, O and Y stand for in the boxes. The caption indicates a specific texture for OM in micropore, but the texture is hard to distinguish from the figure.

In the Figure legend, please recall the name of the model to make the figure understandable on its own.

Comment 4

Line 233: Why were parameters in Table 1 fixed, while only parameters in Table 2 were used in the calibration? Authors: Some reasons were already given at lines 234 to 239. However, we realize now that

this description was insufficient. We will add some new text as well as supporting references in tables 1 and 2 in the revised version to mention these additional reasons. At lines 256-269, the text will be revised to: "). This is because we wanted to critically test the model to see if it was possible to obtain acceptable parameterizations common to all three of the treatments. Inspection of the model equations led us to expect to encounter significant equifinality. Therefore, only six of the fifteen parameters were included in the GLUE analysis, with their prior uncertainty ranges shown in Table 1. The OM supply prior to the start of the experiment and the fraction of this OM supplied as straw, were included in the calibration process to help initialize the SOM pools during a common 5000-year spin-up period. Four remaining parameters, which were considered difficult to identify "a priori" from experimentation, but which were expected to be sensitive and therefore potentially identifiable by calibration, were treated as uncertain (Table 1). We ran 12000 simulations using Latin Hypercube Sampling to sample uniform distributions between the minimum and maximum values for the six uncertain parameters (Table 1). The remaining nine parameters were set to fixed values (Table 2) as they could be estimated from measurements (e.g. fclay, fagg, Fp) or they were not expected to be sensitive (e.g. ky, Andrén and Kätterer, 1997; Juston et al., 2010; Meurer et al., 2020), or both (e.g. $\Psi_{ae}$, $\Psi_{mic}$, $\Theta_{min}$, $\lambda_o$, $\lambda_m$). ".

Reviewer: The text is quite clear about the process of reducing complexity (fixed parameters). Nevertheless, it would be useful to list the four uncertain parameters in the text l 261-263. Also please write "The four remaining parameters" instead of "Four remaining parameters" at line 261

Comment 5

Line 253-254: Why was the mean model efficiency (EF) across all three treatments used to identify acceptable parameter sets?
Authors: Because we wanted to obtain a common parametrization for all three treatments. We will clarify this in the revised version. We will add text at Lines 290-292: "This was done in order to obtain a robust parameterization by selecting parameter sets that simultaneously fitted all three treatments well. The number of acceptable parameter sets was determined such that the range of variation of their predictions approximately covered the variations observed in the measurements. "

Reviewer: The overall explanation of the model parametrization makes it clear that a shared parametrization leads to a reduction of equifinality issues.

- Does this mean that the same parameter set was used for all treatments after calibration?
Authors: Yes. This was implied earlier at lines 231 to 232. However, in the revised version, we will clarify this by writing explicitly that we wanted to obtain a common parametrization for all three treatments.

Reviewer: ok

-     Why not use treatment-specific parameter sets?

Authors: This is as stated above to obtain a robust model parameterization of the model, i.e. parameters sets that should be valid for a wide range of conditions. This is also a way to reduce issues with parameter equifinality.

Also, if parameter values must be changed to account for different treatments (in this case, the amount and type of organic matter inputs), it is a sure sign that something important is missing or wrong in the model, with respect to process descriptions. This means, in turn, that predictions made for contrasting conditions (re. OM inputs) using those calibrated parameter sets may be wildly wrong.

We wanted to critically test the model to see whether it could match the data from the three treatments with a common parameterization. It passed this test.

Reviewer: ok

- Wouldn't taking the mean EF lose treatment-specific information that could be valuable for refining                                                    the                                                    model?
Authors: In principle, yes, it would, but only if the model had performed poorly. However, as shown in figure 2, and as we wrote at lines 308 to 310 (and in the abstract at lines 15 to 20), we were able to get excellent calibrated fits to the data from the three treatments with exactly the same parameterization. This demonstration of the capability of such a simple model is an important result of this study.

Reviewer: ok

Comment 6

Line              257:              15              model              parameters              (Table              3?)
Authors: Yes, the 15 parameters in table 3 were included in the sensitivity analysis.

Reviewer: Several problems

- it is really hard to understand the correspondance between parameters involved in parameters estimation (Table 1 et 2) and sensitivity analysis (Table 3). Please indicate which parameters from Table 2 you considered fixed for the whole region PO4 and were not included in the sensitivity analysis, and why you considered them fixed?

- why Psy_mic values are not consistent between table 2 and 3?

We now refer to Table 3 in the text, Line 299. Line 320, 329,332, 346, 352: Graphs texts are too              small              and              difficult              to              read.
Authors: OK, yes, we made revised versions of these figures with more legible text

REVIEWER: ok

Comment 7

Line 314: Given that most parameters were not well constrained, could parameter covariance be              a              model              artifact              or              a              coincidence?
Authors: It is a consequence of the model structure, so not a coincidence. We feel that this is already stated quite clearly at lines 314 to 316.  (note that we would not like to call this a model

artifact, because it is inherent to the model and not something that occurs by chance as a consequence of the calibration methods applied)

REVIEWER: I agree with the fact that the structure of the model implies correlation between parameters. I would not name this as a model artefact.

- Could the current dataset be insufficient to constrain these parameters?
Authors: This is a good point: to some extent perhaps. For example, if we had the same kind of data from tilled and untilled soils (so 6 treatments at the site, three different OM inputs, with and without tillage), then we would probably have been able to more clearly identify the parameter ktill but only under the condition that the model describe the effects of tillage in a reasonable way (which we don't know yet).

REVIEWER: yes

- Would incorporating additional data sources (e.g., isotope data, incubation experiments) help resolve this issue? If yes, how can this modeling work be robust?
Authors: We mentioned in the introduction (L55-58) that in situ isotope data is also a possible way to reduce some of the model parameter uncertainty, under the condition that they would be prove to be sensitive. We are more doubtful about incubation data based on disturbed/sieved soil samples. We think that the approach adopted here is anyway rather robust, as it included data from several treatments simultaneously.

It is also worth noting that this is the first application of a new model. We wrote at line 404 in the section "Concluding remarks" that the tests of the model in this paper suggest that it "shows promise". We don't think this claim is unreasonably strong. And as we wrote at lines 407 to 409, a greater degree of confidence in the robustness of the model can, of course, be established over time by showing that it produces acceptable results when repeatedly tested against different data sets.

REVIEWER: ok.

Comment 8

Line 316: Why were only the 30 best parameter sets selected?
Authors: Because their predictions were sufficient to cover the range of variability observed in the measurements which is a criteria for the GLUE method. This was implied in the text at lines 291 to 293, but it was not explained. We will mention this criteria in the M&M section after lines 289-294. "This was done in order to obtain a robust parameterization by selecting parameter sets that simultaneously fitted all three treatments well. The number of acceptable parameter sets was determined such that the range of variation of their predictions approximately covered the variations observed in the measurements. With this criterion, 30 of the 12000 parameter sets were identified as acceptable. Note that this low acceptance rate is a consequence of the inefficient sampling inherent to the GLUE method and says nothing about the quality of the model."

REVIEWER: ok

- What was the acceptance rate of parameter sets out of 12,000 simulations?
Authors: 30 out of 12000 = 0.25%. This small value is a consequence of the inefficient sampling which is inherent in the GLUE method: note that it says nothing about the quality of the model. We can state this in a follow-up sentence lines 293-294: "Note that this low acceptance rate is a consequence of the inefficient sampling inherent to the GLUE method and says nothing about the quality of the model."

REVIEWER: yes.

Comment 9

Line 317: The phrase "strong correlation" is used, but no statistical analysis (e.g., correlation coefficients, p-values) is provided to support this claim. Including quantitative analysis would strengthen this statement.

Authors: Yes, we will add R2 and p-values in a revised version of the figure (new Figure 5, page 24). All four relationships are highly significant (p<0.002) with R2 values varying between 0.30 and 0.76.

REVIEWER: ok

Comment 10

Line 338: Figures 5 and 6 are difficult to interpret.

- A more detailed explanation of what these figures represent would improve clarity.
- What key insights should the reader take from these figures?
Authors: Yes, we agree that this was not well explained. In addition to what we wrote at lines 336 to 338, these figures also suggest that the results of the sensitivity analysis should be "reasonably well grounded in reality". We wrote this at line 343 in connection with figure 7, but this conclusion should also be based on figures 5 and 6 (now figure 6). In the revised version, we will modify the text at lines 380 to 389 to make this clearer. "A qualitative comparison with soil survey data for agricultural land in east-central Sweden (production area PO4) suggests that despite its simplicity the model estimates of steady-state SOC and bulk density in the soil profile lie mostly within the range of variation encountered in the region (Figure 6). Nevertheless, quantile-quantile plots show that the distributions of simulated and measured values of SOC and bulk density are different; especially at the tails, due to the much larger spread in the measurements compared with the calculations and especially the occurrence of a number of outliers with large values of organic carbon contents and small values of bulk density. This is not surprising because the calculations do not include the effects of all factors affecting SOC and bulk density. The large values of SOC (and small values of bulk density) almost certainly correspond to locations in the PO4 region with wet soils due to topography (i.e. flood plains, depressions). The model, as it is formulated here, does not include the effects of excess soil moisture on decomposition rates." We also removed the following sentence in the abstract, Line 23-24 ". The resulting model predictions compared well with aggregated soil survey data for the PO4 region" as this can be wrongly interpreted.

REVIEWER: Maybe the authors should propose the q-q plots as supplementary material.

Comment                                                                              11
Line 345: Minor inconsistency: (e.g., consistently use "Figure X" or "Fig. X rather than mixing "Fig.                X"               and               "Figure               X").
Authors: Yes, we fixed this in the revised version, we now use "Figure" at start of a sentence and "Fig." otherwise to match the SOIL journal guidelines.

REVIEWER: ok

Response to comments from Reviewer 2

Authors: We would like to thank the Reviewerfor the positive and perceptive comments, as well as for the questions, which will help us to improve the paper, specifically by giving more details, information and explanations about our data, methods and results.

This study describes a simple model of soil organic carbon (SOC) turnover that represents the effects of soil physical protection and microbial energy limitation. The paper first describes the model in a soil profile, then tests it using SOC data from a long-term study on agricultural fields with varied C inputs to the soils. Finally, the model's most influential parameters were identified in a sensitivity and uncertainty analysis. Overall, the paper is well written and presents a model of interest and relevance to soil carbon management. Below, please find specific comments intended to help improve the paper.

Comment 1

L83 – Define the abbreviation USSF.
Authors: OK, we will do this in the revised version (it stands for Uppsala model of Soil Structure and Function), see line 93.

REVIEWER: ok

Comment 2

L91-95 – I would suggest ending the introduction with a strong thesis statement of what the paper contributes to current knowledge of the subject.
Authors: Yes, we will add such a statement at the end of the introduction. See Lines 104-107 "The overall aim of this study is to demonstrate the utility of a simple soil C turnover model that can account for the nexus of soil management, soil structure and microbial activity that critically determines C mineralization and stabilization at the scale of a soil profile."

REVIEWER: the presentation of the purpose can still be improved, I suggest rephrasing the last paragraph entirely instead of only adding a final sentence. The research gap addressed by the authors should be mentioned at the end of the second last paragraph and connected with the general objectives of the study at the beginning of the last paragraph. The paper is composed of several phases, which are hard to follow. It is necessary to emphasize these phases in this last paragraph by adding "First,...", "second, ..." etc... You could also move the first lines of the M&M section in this last paragraph of introduction (see next comment).

Comment 3

L97-101 – I appreciate this overview of the methods, very helpful to have this framework.
Authors: Thanks!

REVIEWER: ok. This overview (line 109-113) could be integrated in the last part of the intro for the sake of clarity.

Comment 4

L103 – In section 2.1.1 that starts on this line, it is unclear if the model as described in this section is the work of the authors or if this is describing previously published work. If it has been previously published, I suggest including most of the equations in this section in a supplement rather than in the main document. In the main manuscript, I suggest describing the model in writing and including important equations for the modifications to the model that are new in the current study.
Authors: The model is based on a combination of the model described by Meurer et al. (2020) accounting for physical protection in relation to soil properties with the model described in Wutzler and Reichstein (2013) for microbial energy limitation. This combination of the two models was outlined in Coucheney et al. (2024), along with some minor improvements and modifications. However, the description of this SOM model was only included in the supplementary information in Coucheney et al. (2024) as the model itself was not tested at all. So although the main constituent components of the model have been described earlier, this is the first time that the complete model has been tested. As the model is quite new and previously untested, we prefer to keep these equations in the main paper, as it makes it easier for the reader. The paper is not too long and including the equations in the main text will ensure that the equations are readily available. We think this history of the model development is clearly explained at lines 104 to 115.

REVIEWER: I agree with the authors. In addition, presentation of equations helps to get the structure of the model which is critical for discussing parameters structural correlations.

Additionally, I would encourage the authors to post their full model code online and cite it in the paper.
Authors: We built the model using the icon-based modelling software STELLA, which is a commercial product. The model file will be made available on request to the authors – this will be stated in the "data availability statement" at the end of the manuscript.

REVIEWER: ok

Comment 5

107-109 – It would be helpful to specify the direction of the relationship between these effects (e.g., Do smaller pores get fewer root derived inputs?)

Authors: Not necessarily, no. It depends on the pore size distribution in the soil, which in turn depends on the soil texture (or more simply clay content in our approach). The pore size distribution determines the partitioning of root-derived inputs of OM between the two pore regions (see Eqs. 10-13). This means that a larger proportion of the root C inputs would enter the micropore region in a clay soil than in a sandy soil, because the porosity of a clay soil predominantly consists of smaller pores. Clay soils therefore have a higher potential for physical protection of soil C. The effect of the physical protection is quantified by the factor $F_p$ (Eqs. 1-4) that reduces the rate of SOC decomposition in micropores.

We will modify the text at Lines 119-124 to make this clearer: "In turn, the pore size distribution determines the partitioning of root-derived inputs of OM between the two pore regions. This means that compared with a sandy soil, a larger proportion of the root OM input will enter the micropore region in a clay soil, as it predominantly consists of smaller pores. The soil pore size distribution also regulates decomposition rates with slower decomposition rates of OM stored in microporous regions of the soil. Compared with sandy soils, clay soils therefore have a greater potential for physical protection of soil C.".

REVIEWER: ok

Do micropores have lower decomposition rates?).
Authors: Yes, they do. We will clarify this in the revised text at lines 107-109 (see above text)

REVIEWER: ok

Comment 6

L125 – Is the "(-)" after fr,mic supposed to indicate that it is unitless?
Authors: Yes

REVIEWER: ok

Comment 7

L218 – How does this straw addition rate compare to the maize biomass per hectare?
Authors: Note that maize was only grown since 2000. Between 2000-2019, the crop fertilized and crop fertilized+straw treatments had on average 6.07 and 7.09 t ha-1 maize yield, respectively (see Kätterer & Bolinder, 2024). However, above-ground maize and crop residues were removed from the field and it is very speculative how much C is added to the soil system via root inputs (rhizodeposition and exudates) and such C inputs are challenging to quantify and are therefore not included in the present proposed model. For the modelling, we know how much straw was added and this is included in the paper.

REVIEWER: ok

Comment 8

L219 - 220 – Include the scientific names/varieties of the crops.
Authors: We will include Latin names in the revised version (lines 242-244): "Maize (Zea mays) has been grown on the cropped plots since 2000. Before 2000, the crop rotation included barley (Hordeum vulgare), oats (Avena sativa), beets (Beta vulgaris) (prior to 1967) and rape (Brassica napus)."

In addition, we propose to include a reference to a newly published data paper which gives a complete description of the experiment, as well as links to all the data (Data in Brief paper. Pold et al. 2025 - Soil and vegetation property data from the Ultuna R3-RAM56 long-term soil amendment experiment, 1956-2023), at line 233.

Comment 9

L220 - What is the purpose of hand digging the plots after harvest?
Authors: It's to simulate ploughing: the plot size of 4 m2 is too small to manage in the same way as a farmer's field. We will add this information at Line 244: "to simulate ploughing as the plots are too small (4 m2) to be managed in the same way as a farmer's field."

Comment 10

L 231 – What measurements were used for the calibration? Only OM or additional measurements?
Authors: Only SOC. We will state this explicitly in the revised version at line 254.

Calibrating to additional variables could help reduce equifinality.
Authors: Perhaps, depending on the type of data and its quantity and quality. But we were able to strongly reduce the prior uncertainty ranges for the parameter values anyway, despite parameter correlation.   See also our answer to comment number 7 from Reviewer1

Comment 11

L238 – Specify the field bulk density values used for this validation.
Authors: OK, yes, we will do so (the numbers are 1.43, 1.28 and 1.21 g/cm3 in the treatments "Fallow", "N-fertilized" and "N fertilized + straw" respectively). This was added at lines 276-277.

Comment 12

Table 1 – Are there field measurements available for any of these parameters? If so, how similar are they to these fixed values?

Authors: Yes, we didn't write it explicitly but these values in table 1 were based (with one exception) on data obtained at the study site. We will add a column to this table with the heading "Source" where we will cite the relevant studies.

See also answer to comment 4 – Reviewer1

Comment 13

Figure 1 needs a legend to identify what the different colors/patterns of shading indicates.
Authors: We will add this information to the caption (at lines 227-229) "Figure 2. Map of Sweden (in white) showing the location of the Ultuna Long-term Soil Organic Matter Experiment (Uppsala, Sweden) and the extent of the production area PO4 (shaded area in grey). Drawn by Anna Lindahl, SLU, from Esri, TomTom, Garmin, FAO, NOAA, and USGS"

REVIEWER: ok. What does PO4 stand for? You could indicate it from the beginning.

Comment 14

Table 1 – Are there data references for these parameter values? If not, how did you come to these values?

Authors: Yes, we will give these references. We will add a column to this table with the heading "Source"

REVIEWER: The question was related to Table 1, not to Table 2. Could you please explain also for Table 1 how you come to these values?

Tables – I recommend including captions for tables with relevant details.
Authors: We will revise the table captions giving more informative detail. For example:

Lines 278-279: "Table 1. Six model parameters selected Initial parameter uncertainty ranges for the model calibration to the Ultuna Long-Term Soil Organic Matter Experiment and their initial parameter uncertainty ranges"

Lines 281-282: "Table 2. Nine model parameters fixed at constant values during the calibration based on field measurements at Ultuna or literature data"

Page 15: "Table 3. Parameter input distributions in the sensitivity analysis. In the case of uniform distributions, minimal and maximal values are shown (Min.; Max.) while in the case of normal distribution the mean and standard deviation are shown (Mean; St. dev.)."

REVIEWER: ok

Comment                                                                    15
L264-    What    determined    if    the    data    support    was    sufficient?
Authors: This is partially a subjective decision. However, we can revise the sentence to be a bit more specific:

Lines 303-305: "We assumed normal distributions when the data support was considered sufficient to support such a distribution. , while uniform distributions were used otherwise (Table 3) "

REVIEWER: ok

Comment                                                                    16

Table 3 – The source "SCB Statistics Sweden" needs to be more specific. Same comment for source listed as "site data" - where are site data accessible? Year, dataset name, authors etc.

Authors: Yes, we have added these details.

REVIEWER: ok

Comment                                                                                                17

L304 - 306 – Is this distinction of straw going into only mesopores and root OM going partially into          micropores          supported          by          empirical          data?

Authors: No, it's more a model hypothesis, although based on process understanding. We can't see how digging or ploughing down above-ground crop residues could possibly incorporate these residues into the microporous regions of the soil where pore diameters are less than 5 microns (in this study). In contrast, roots can grow through microporous soil regions supplying both POM (on root death) and root exudates.

This hypothesis could be tested experimentally in future work with for example, the help of X-ray tomography on samples taken soon after harvest and ploughing-down of above-ground crop residues). This is however way beyond the scope of the present study.

REVIEWER: Could you please explain this in the text?

Comment 18

L308 – What is meant by "export of residues?" Does that mean the removal of residues by land managers?

Authors: Yes. In this treatment, most of the above-ground crop residues are removed. This was explained at line 217. We will replace this phrase by , at line 350: "due to the near total removal of above-ground crop residues."

REVIEWER: ok

Comment                                                                                                19

Figure 3 – This panel figure needs letters for each panel and a description of each panel and the          definition          of          the          X          axes          in          the          figure          caption.

Authors: We feel that letters for the panels are not needed but we will give further details in the figure caption instead, page 22 (now figure 4):

"Figure 4. Mean model efficiencies for each parameter set (only simulations with model efficiencies larger than zero are shown) plotted against the values for the six parameters in the GLUE analysis (refer to table 1 for parameter definitions and descriptions; OM = organic matter, AG = above-ground)"

REVIEWER: ok. To make the figure understandable on its own, could you briefly recall how it was generated? L355: Explain for non-GLUE users and non-statisticians why we can graphically conclude from new Fig 4 that epsilon is constrained, while the other parameters are not.

Figure 4 – This panel figure also needs to have the individual panels labeled/described and the axes defined in the caption.

Authors: We feel that letters for the panels are not needed because we don't refer to individual subfigures in the text, but we will give further details in the figure caption instead, page 24 (now figure 5): "Figure 5. Inter-relationships among four of the six model parameters included in the calibration procedure. Relationships are shown for the 30 best parameter sets identified in the GLUE analysis (refer to table 1 for parameter definitions and descriptions)."

REVIEWER: Thank you, but please provide the meaning of the parameters instead of asking to refer to Table 1

Comment 20

L314 – In the introduction, large mechanistic models are criticized for having uncertainty and equifinality. That was provided as justification for a simpler parsimonious model. How does that criticism relate to your finding that the simple/parsimonious model presented here has the same issue of equifinality and parameter uncertainty as the more complex models? How does this affect the usefulness of the model or its applicability compared to the larger models? Or, should this model's parameters be further simplified?

Authors: Please see our answer to comment 2 of Reviewer1 (also copied here) Yes, this is a good point. It's true that even the simplest models of organic matter turnover in soil can show equifinality, depending on the type, quantity and quality of the data used to constrain them. This has been demonstrated for models that are even simpler than the one we developed and tested in our study (see e.g. Juston et al., 2010; Luo et al., 2017). We were therefore expecting to encounter the issue, as we wrote at line 233. This is also why we wrote "may" on line 48.

However, in contrast to more complex models, we can clearly see why and where the equifinality arises in our relatively simple model: it depends on the model structure, with correlations among only a few parameters, which makes the problem of parameter uncertainty more manageable. This was the case for the model application to the data at the Ultuna long-term soil organic matter experiment presented in the paper: here, simultaneous calibration to data was sufficient to effectively constrain the model parameters in three treatments with strongly contrasting inputs of OM with respect to both type and amount. This is what we concluded at lines 317-318: "These strong correlations of ko, Aa and ktill with epsilon mean that, in practice, all four parameters are well constrained by the calibration".

REVIEWER: ok

Comment                                                                                                          21
L317 – What is the evidence of strong correlation? This statement needs statistical support.
Authors: Yes, we now added R2 and p-values in a revised version of the new figure 5. All four relationships are highly significant (p<0.002) with R2 values varying between 0.30 and 0.76.

REVIEWER: ok

Comment                                                                                                          22

L341         –         Temperate,         not         temperature.
Authors: Thanks, we fixed this

REVIEWER: ok

Comment 23

Figure 5 – Can you provide a quantitative comparison of the means to support the conclusions?
Authors: This is a good point. We didn't attempt any statistics, simply writing that the model gives reasonably realistic predictions". In fact, statistical tests for differences between the two distributions (not just the means) show that they are significantly different, which is almost entirely due to the much larger spread in the measurements compared to the calculations, especially the occurrence of a number of outliers with large values of organic carbon contents and correspondingly small values of bulk density. This is not really surprising because the calculations do not include the effects of all factors affecting SOC and bulk density. Our guess is that the large values of organic carbon content (and small values of bulk density) correspond to locations with wet soils due to topography (i.e. flood plains, depressions). The model, as it is formulated here, does not include the effects of excess soil moisture on decomposition rates.

We have added some new text in the revised version at lines 380-389 to explain the above. "A qualitative comparison with soil survey data for agricultural land in east-central Sweden (production area PO4) suggests that despite its simplicity the model estimates of steady-state SOC and bulk density in the soil profile lie mostly within the range of variation encountered in the region (Figure.s 6 and 7). Nevertheless, quantile-quantile plots show that the distributions of simulated and measured values of SOC and bulk density are different; especially at the tails, due to the much larger spread in the measurements compared with the calculations and especially the occurrence of a number of outliers with large values of organic carbon contents and small values of bulk density. This is not surprising because the calculations do not include the effects of all factors affecting SOC and bulk density. The large values of SOC (and small values of bulk density) almost certainly correspond to locations in the PO4 region with wet soils due to topography (i.e. flood plains, depressions). The model, as it is formulated here, does not include the effects of excess soil moisture on decomposition rates."

We will avoid using the word "predictions", because these are the aggregated outputs of a sensitivity analysis and not model predictions that can be compared with measurements at specific locations.

REVIEWER:         Please         add         q-q         plots         as         supplementary

Comment 24

Figure 6 – Why are only two depths shown in this figure? (there are 3 depths in the previous figure)
Authors: Because we only have bulk density data at two depths, i.e. between 0-20 cm depth and 40-60 cm depth, whereas SOC was measured between 0-20, 20-40 and 40-60 cm depth (see line 281). This appears better now that both figures have been merged under the name Figure 6 (see page 25)

REVIEWER: ok

L357 – What are the cutoffs for NRC values to determine if they are strongly, moderately, or minimally sensitive in this analysis? I think that information should be included in the methods. In the discussion, it would be helpful to more quantitatively compare/describe the model sensitivity to these various parameters.

Authors: We prefer to discuss these results in relative terms and therefore we choose to present all values ordered decreasingly in Table 5 without any use of defining cut off values that may be judged to be arbitrary (the readers can also see all of the figures and apply their own criteria if wanted)

REVIEWER: the current version of the paper is understandable; however, it is necessary to indicate the ranking method in the table legend.

Comment 25

L359 – The fraction of aboveground residues incorporated is roughly as important as the clay content yet it is not mentioned here.

Authors: Yes, we agree that it should also be mentioned. We will include a mention of finc at line 418, alongside the other two parameters that determine the input of above-ground residues.

REVIEWER: ok

I'm also unclear why the clay content is mentioned before the other more sensitive parameters in the table.

Authors: Yes, we see your point. We mentioned the clay content here because it belongs together with Fp in that taken together they determine the extent of physical protection. We did try to explain this at lines 414-416, but we will make this connection much clearer in the revised version:

"The soil clay content, which together with Fp, determines the extent to which physical protection is expressed in soils of contrasting texture, is also a relatively sensitive model parameter (Table 5)"

REVIEWER: ok

Comment 26

L367-370 – I would be interested to see this idea expanded upon – how does this USSF model result relate back to your model result of an 8% increase in SOM?

Authors: It's really mostly just a consequence of the fact that it takes a lot longer than 30 years to reach steady-state after a change in OM inputs. We added the word "shorter" to highlight this at line 424.

REVIEWER: ok

And what are the larger implications of these increases for climate change mitigation as you mention? For example, how does a 1.4% increase over the course of 30 years compare to targeted goals for mitigation?

Authors: This is discussed in Coucheney et al. (2024) and we refer to this study in the text. We don't think this is the right place to discuss the results from a previous paper.

Comment 27

L379 – Here, the authors seem to consider a 4-5% reduction in SOM to be minor. But on lines 374 - 377 they seem to indicate that a 3-5% increase is significant. Some benchmarks for the relevance of these changes would be helpful for interpretation of the results. Authors: Thanks for pointing this out. It is only an apparent inconsistency, because the 3-5% increase mentioned at lines 374-377 was after 30 years, whereas at line 379 we are referring to steady-state values. We will make this clearer by slightly modifying the sentence at L433-434: "Table 5 suggests that tillage is one of the least sensitive factors affecting SOM stocks at steady- state: "

Comment 28

L381 – Haddaway et al. found a difference by tillage intensity in the topsoil, as opposed to what is stated here. Intermediate intensity tillage resulted in greater SOC stocks than the high intensity tillage in that metanalysis. Authors: This seems to be a misunderstanding. We did write that Haddaway et al. found greater stocks of C in the topsoil under no-till. However, we also wrote that they did not find any difference in total C stocks in the profile between no-till and intensive or intermediate tillage systems, which is also true. We will slightly revise the sentence at lines 437-440: "…. Haddaway et al. (2017) and Meurer et al. (2018) only found larger SOC stocks under no-till compared with conventional tillage in the topsoil, but significant differences in total SOC stocks in these two tillage systems in soil profiles to 60 cm depth."

Comment 29

L385 – It would be helpful to include empirical data supporting these model data on the figure for comparison with the model data. Or include the empirical data in a table caption. Authors: We would prefer not to modify the actual figure to show data from other studies. This is partly because a direct comparison of predictions for a region in east-central Sweden with measurements from a highly diverse global dataset like Chen et al. (2020) could give readers a misleading impression. In addition, the raw data on MRT from the study by Poeplau et al. (2021) are not available to us. The value quoted in the paper at line 339, "ca. 20 years" was taken from a table in the paper (table 1). In the revised version of the paper, we will also add the values reported by Chen et al. (2020) for cropland to this text, as this data is available. In doing so, we will also slightly modify the text at lines 395 to 399: "…. and they also lie at the high end of the range in the global analysis reported by Chen et al. (2020) for croplands (mean = 9.5 years, standard deviation = 6 years, n = 217). Taken together with Fig. 6, this gives us confidence that the results of the sensitivity analysis presented in the following should be reasonably well grounded in reality." We feel this is better than adding these numbers to the figure caption.

Comment 30

Table 5 – It looks like this table was color coded according to the colors of the groups in table 3. That information and the meaning of the colors should be included in the caption. Authors: Yes, that's right. However, we have been advised by the editor that we must remove this colour coding in the revised version of the paper.

REVIEWER: ok

References cited (in answers to both Reviewer1 and 2)

Andrén, O., Kätterer, T. 1997. ICBM: the introductory carbon balance model for exploration of soil carbon balances. Ecological Applications, 7, 1226-1236.

Chen, S., Zou, J., Hu, Z., Lu, Y. 2020. Temporal and spatial variations in the mean residence time of soil organic carbon and their relationship with climatic, soil and vegetation drivers. Global and Planetary Change, 195, 103359.

Coucheney, E., Kätterer, T., Meurer, K.H.E., Jarvis, N. 2024. Improving the sustainability of arable cropping systems by modifying root traits: a modelling study for winter wheat. European Journal of Soil Science, 75, e13524.

Dechow, R., Franko, U., Kätterer, T., Kolbe, H. 2019. Evaluation of the RothC model as a prognostic tool for the prediction of SOC trends in response to management practices on arable land. Geoderma, 337, 463-478.

Kätterer, T., Bolinder, M. 2024. Response of maize yield to changes in soil organic matter in a Swedish long-term experiment. European Journal of Soil Science, 75, e13482.

Juston, J., Andrén, O., Kätterer, T., Jansson, P-E. 2010. Uncertainty analyses for calibrating a soil carbon balance model to agricultural field trial data in Sweden and Kenya. Ecological Modelling, 221, 1880-1888.

Luo, Z., Wang, E., Sun, O. 2017. Uncertain future soil carbon dynamics under global change predicted by models constrained by total carbon measurements. Ecological Applications, 27, 1001-1009.

Meurer, K., Chenu, C., Coucheney, E., Herrmann, A., Keller, T., Kätterer, T., Nimblad Svensson, D., Jarvis, N. 2020. Modelling dynamic interactions between soil structure and the storage and turnover of soil organic matter. Biogeosciences, 17, 5025-5042

Poeplau, C., Don, A., Schneider, F. 2021. Roots are key to increasing the mean residence time of organic carbon entering temperate agricultural soils. Global Change Biology, 27, 4921–4934.

Pold, G., MacDonald, E., Braun S., Herrmann, A. M., 2025. Soil and vegetation property data from the Ultuna R3-RAM56 long-term soil amendment experiment, 1956-2023

Wutzler, T., Reichstein, M. 2013. Priming and substrate quality interactions in soil organic matter models. Biogeosciences, 10, 2089-2103.

---

## Author Response (AR2)

Dear Editor of the SOIL journal,

Please find our answers to the third reviewer comments (in green), as well as a revised manuscript according to all points raised by the reviewer. This includes minor updates on the text and figure captions, a rewriting of the aim of the paper in the introduction, as well as an additional figure to complete Figure 6 in an appendix (i.e. Figure A1, Q-Qplots of the regional sensitivity analysis) and an additional reference (Balesdent, 2018).

Dear Editor, please find below a check that the referees' comments have been incorporated satisfactorily in the final revised version of the article submitted by Coucheney et al, entitled "A simple model of the turnover of organic carbon in a soil profile: model test, parameter identification and sensitivity"
The referees' comments are in black, the authors' responses in blue, my comments in red.
* * *
Dear Editors of the SOIL journal,

We hereby present the revised manuscript entitled "A simple model of the turnover of organic carbon in a soil profile: model test, parameter identification and sensitivity", in which we have implemented revisions according to the suggestions and questions of the two reviewers.

This includes

- The inclusion of additional text in the introduction, results and discussion sections to further clarify or specify our approach (for example, how we deal with the equifinality issue in our modelling).

- An additional figure (Figure 1) showing a schematic of the model developed and applied.

- A review of the quality of all figures (and colour changes where appropriate) and table captions (with more specific text).

- In addition, note that former Figures 6 and 7 have been combined into a single figure (now Figure 6), so that Figure 8 has become Figure 7.

- References have been updated and completed. In addition, the format for publication in SOIL has been updated.

- A data availability and model statement has been added at the end of the manuscript.

Below is a detailed description of the revision in response to the Reviewer comments. Line references are to the tracked-changes document (PDF) I hope you find this manuscript suitable for publication in SOIL.

Best regards,
Elsa Coucheney, for the co-authors

Response to comments from Reviewer1

Authors:
We would like to thank the Reviewer for the positive and perceptive comments, as well as for the questions, which will help us to improve the paper, specifically by giving more details, information and explanations about our data, methods and results.

Reviewer:
The study introduces a parsimonious soil organic carbon (SOC) turnover model for the soil profile that includes key processes controlling carbon persistence. The model specifically incorporates two crucial mechanisms often omitted in simpler models: (1) microbial energy limitation (a form of positive/negative priming where decomposition slows if substrate is scarce) and (2) physical protection via soil aggregation (which protects organic matter from decomposition). The aim was to test this model against long-term field data, identify how well model parameters can be determined (parameter identifiability), and analyze sensitivity of the model to its parameters. However, most of the parameters were not constrained and covariance between the parameters were used as an explanation to parameter unidentifiable. The manuscript is well-written and please find the specific comments as follows:

**Comment 1**

Line 34, 36: What soil depths represent topsoil and subsoil? Providing the depth ranges would be helpful for clarity.

Authors: Here, we define the topsoil as the cultivated (tilled) soil layer, rather than in terms of depth ranges. We will clarify this in the revised version L33, by writing: "Most model applications to date have focused on the cultivated topsoil, which is clearly of major importance with respect to the effects of soil management on SOC and soil health"

REVIEWER: the notion of depths remains unprecise but at this point of the introduction, the proposed change is sufficient to figure out model applications.

OK

**Comment 2**

Line 45-49: The study demonstrated parameter uncertainty and equifinality, despite having fewer parameters than complex models. However, this raises the question: how is the simple model used in this study different from detailed mechanistic models if parameters remain unconstrained? An explanation would help readers understand the trade-offs between complexity and parameter uncertainty.

Authors: Yes, this is a good point. It's true that even the simplest models of organic matter turnover in soil can show equifinality, depending on the type, quantity and quality of the data used to constrain them. This has been demonstrated for models that are even simpler than the one we developed and tested in our study (see e.g. Juston et al., 2010; Luo et al., 2017). We were therefore expecting to encounter the issue, as we wrote at line 233. This is also why we wrote "may" on line 48.

However, in contrast to more complex models, we can clearly see why and where the equifinality arises in our relatively simple model: it depends on the model structure, with correlations among only a few parameters, which makes the problem of parameter uncertainty more manageable. This was the case for the model application to the data at the Ultuna long-term soil organic matter experiment presented in the paper: here, simultaneous calibration to data was sufficient to effectively constrain the model parameters in three treatments with strongly contrasting inputs of OM with respect to both type and amount. This is what we concluded at lines 317-318: "These strong correlations of $k_o$, $A_a$ and $k_{till}$ with epsilon mean that, in practice, all four parameters are well constrained by the calibration".

In the revised version, we will clarify these issues by revising and adding to the text at lines 52 to 61: " This is because these models have often been tested against limited datasets (i.e. observations of topsoil C dynamics at a single site and treatment) which increases the likelihood of equifinality despite the small number of parameters (e.g. Juston et al., 2010; Luo et al., 2017). This may be overcome by simultaneous calibration of the model against data for two or more contrasting treatments, for example with respect to the type and quantity of organic matter inputs (e.g. Meurer et al., 2020) or by multi-site calibration at larger scales using data from long-term field trials at locations with contrasting soils and management practices (e.g. Juston et al., 2010; Dechow et al., 2019). Testing model predictions for entire soil profiles remains however difficult and is therefore rarely done, because fewer measurements are made in subsoils and the turnover of organic C in subsoil is very slow, so datasets will rarely be long enough to detect any changes. Additional data sources may also help to alleviate problems arising from equifinality. … "

Reviewer: Generally, the manuscript is now clear concerning equifinality and analysis of parameters obtained via the GLUE procedure, constraints, and correlations.

OK

**Comment 3**

Line 70: What does ICBM stand for? Additionally, the reference to Andrén and Kätterer (1997) is missing. A schematic diagram illustrating model development over time would help readers visualize how the model has evolved. Similarly, a conceptual diagram of the final model used in this study would be beneficial alongside the mathematical equations.

Authors: ICBM stands for Introductory Carbon Balance Model. We apologize for not including the reference to Andrén and Kätterer (1997). We will add this to the revised version in line 79. We will also include a conceptual diagram of the model to go alongside the equations which is referred to Figure 1, page 5.

Reviewer: Please explain what do M, O and Y stand for in the boxes.

OK, we have done so.

The caption indicates a specific texture for OM in micropore, but the texture is hard to distinguish from the figure.

Clay content determines the proportions of micropores and mesopores in a soil, which in turn controls the partitioning of root-derived OM between them. This is stated in the caption (and in the model description) and is shown in the bottom right-hand corner of the figure.

In the Figure legend, please recall the name of the model to make the figure understandable on its own.
We can't do this because we do not name the model anywhere in the paper. In the caption, we help the readers by referring to the relevant equations for each part of the model.

**Comment 4**

Line 233: Why were parameters in Table 1 fixed, while only parameters in Table 2 were used in the calibration?

Authors: Some reasons were already given at lines 234 to 239. However, we realize now that this description was insufficient. We will add some new text as well as supporting references in tables 1 and 2 in the revised version to mention these additional reasons. At lines 256-269, the text will be revised to: "). This is because we wanted to critically test the model to see if it was possible to obtain acceptable parameterizations common to all three of the treatments. Inspection of the model equations led us to expect to encounter significant equifinality. Therefore, only six of the fifteen parameters were included in the GLUE analysis, with their prior uncertainty ranges shown in Table 1. The OM supply prior to the start of the experiment and the fraction of this OM supplied as straw, were included in the calibration process to help initialize the SOM pools during a common 5000-year spin-up period. Four remaining parameters, which were considered difficult to identify "a priori" from experimentation, but which were expected to be sensitive and therefore potentially identifiable by calibration, were treated as uncertain (Table 1). We ran 12000 simulations using Latin Hypercube Sampling to sample uniform distributions between the minimum and maximum values for the six uncertain parameters (Table 1). The remaining nine parameters were set to fixed values (Table 2) as they could be estimated from measurements (e.g. $f_{clay}$, $f_{agg}$, $F_p$) or they were not expected to be sensitive (e.g. $k_y$, Andrén and Kätterer, 1997; Juston et al., 2010; Meurer et al., 2020), or both (e.g. $\Psi_{ae}$, $\Psi_{mic}$, $\Theta_{min}$, $\lambda_o$, $\lambda_m$). ".

Reviewer: The text is quite clear about the process of reducing complexity (fixed parameters). Nevertheless, it would be useful to list the four uncertain parameters in the text l 261-263. Also please write "The four remaining parameters" instead of "Four remaining parameters" at line 261
OK, yes, we have changed to … "The four other parameters …" as this makes it even clearer what we mean.

**Comment 5**

Line 253-254: Why was the mean model efficiency (EF) across all three treatments used to identify acceptable parameter sets? Authors: Because we wanted to obtain a common parametrization for all three treatments. We will clarify this in the revised version. We will add text at Lines 290-292: "This was done in order to obtain a robust parameterization by selecting parameter sets that simultaneously fitted all three treatments well. The number of acceptable parameter sets was determined such that the range of variation of their predictions approximately covered the variations observed in the measurements. "

Reviewer: The overall explanation of the model parametrization makes it clear that a shared parametrization leads to a reduction of equifinality issues.

OK

- Does this mean that the same parameter set was used for all treatments after calibration?

Authors: Yes. This was implied earlier at lines 231 to 232. However, in the revised version, we will clarify this by writing explicitly that we wanted to obtain a common parametrization for all three treatments.

Reviewer: ok

OK

-    Why not use treatment-specific parameter sets?

Authors: This is as stated above to obtain a robust model parameterization of the model, i.e. parameters sets that should be valid for a wide range of conditions. This is also a way to reduce issues with parameter equifinality. Also, if parameter values must be changed to account for different treatments (in this case, the amount and type of organic matter inputs), it is a sure sign that something important is missing or wrong in the model, with respect to process descriptions. This means, in turn, that predictions made for contrasting conditions (re. OM inputs) using those calibrated parameter sets may be wildly wrong.

We wanted to critically test the model to see whether it could match the data from the three treatments with a common parameterization. It passed this test.

Reviewer: ok
OK

- Wouldn't taking the mean EF lose treatment-specific information that could be valuable for refining the model?

Authors: In principle, yes, it would, but only if the model had performed poorly. However, as shown in figure 2, and as we wrote at lines 308 to 310 (and in the abstract at lines 15 to 20), we were able to get excellent calibrated fits to the data from the three treatments with exactly the same parameterization. This demonstration of the capability of such a simple model is an important result of this study.

Reviewer: ok

OK

**Comment 6**

Line 257: 15 model parameters (Table 3?)
Authors: Yes, the 15 parameters in table 3 were included in the sensitivity analysis.

Reviewer: Several problems

- it is really hard to understand the correspondance between parameters involved in parameters estimation (Table 1 et 2) and sensitivity analysis (Table 3). Please indicate which parameters from Table 2 you considered fixed for the whole region PO4 and were not included in the sensitivity analysis, and why you considered them fixed?

Only one of the parameters in table 2 was considered fixed for the whole region (the pressure head defining the upper pore size limit for mesopores). This is because there is no 'a priori' physical reason to expect it should vary among different soils. We have added a sentence to explain this

- why Psy_mic values are not consistent between table 2 and 3?

A minus sign was missing in table 2. We have fixed this
Line 320, 329,332, 346, 352: Graphs texts are too small and difficult to read.
 Authors: OK, yes, we made revised versions of these figures with more legible text

REVIEWER: ok
OK

**Comment 7**

Line 314: Given that most parameters were not well constrained, could parameter covariance be a model artifact or a coincidence?

Authors: It is a consequence of the model structure, so not a coincidence. We feel that this is already stated quite clearly at lines 314 to 316.  (note that we would not like to call this a model artifact, because it is inherent to the model and not something that occurs by chance as a consequence of the calibration methods applied)

REVIEWER: I agree with the fact that the structure of the model implies correlation between parameters. I would not name this as a model artefact.
OK

-  Could the current dataset be insufficient to constrain these parameters?

Authors: This is a good point: to some extent perhaps. For example, if we had the same kind of data from tilled and untilled soils (so 6 treatments at the site, three different OM inputs, with and without tillage), then we would probably have been able to more clearly identify the parameter ktill but only under the condition that the model describe the effects of tillage in a reasonable way (which we don't know yet).

REVIEWER: yes
OK

- Would incorporating additional data sources (e.g., isotope data, incubation experiments) help resolve this issue? If yes, how can this modeling work be robust?

Authors: We mentioned in the introduction (L55-58) that in situ isotope data is also a possible way to reduce some of the model parameter uncertainty, under the condition that they would be prove to be sensitive. We are more doubtful about incubation data based on disturbed/sieved soil samples. We think that the approach adopted here is anyway rather robust, as it included data from several treatments simultaneously.

It is also worth noting that this is the first application of a new model. We wrote at line 404 in the section "Concluding remarks" that the tests of the model in this paper suggest that it "shows promise". We don't think this claim is unreasonably strong. And as we wrote at lines 407 to 409, a greater degree of confidence in the robustness of the model can, of course, be established over time by showing that it produces acceptable results when repeatedly tested against different data sets.

REVIEWER: ok.

OK

**Comment 8**

Line 316: Why were only the 30 best parameter sets selected?

Authors: Because their predictions were sufficient to cover the range of variability observed in the measurements which is a criteria for the GLUE method. This was implied in the text at lines 291 to 293, but it was not explained. We will mention this criteria in the M&M section after lines 289-294. "This was done in order to obtain a robust parameterization by selecting parameter sets that simultaneously fitted all three treatments well. The number of acceptable parameter sets was determined such that the range of variation of their predictions approximately covered the variations observed in the measurements. With this criterion, 30 of the 12000 parameter sets were identified as acceptable. Note that this low acceptance rate is a consequence of the inefficient sampling inherent to the GLUE method and says nothing about the quality of the model."

REVIEWER: ok

OK

What was the acceptance rate of parameter sets out of 12,000 simulations?
Authors: 30 out of 12000 = 0.25%. This small value is a consequence of the inefficient sampling which is inherent in the GLUE method: note that it says nothing about the quality of the model. We can state this in a follow-up sentence lines 293-294: "Note that this low acceptance rate is a consequence of the inefficient sampling inherent to the GLUE method and says nothing about the quality of the model."

REVIEWER: yes.
OK

**Comment 9**

Line 317: The phrase "strong correlation" is used, but no statistical analysis (e.g., correlation coefficients, p-values) is provided to support this claim. Including quantitative analysis would strengthen this statement.

Authors: Yes, we will add R2 and p-values in a revised version of the figure (new Figure 5, page 24). All four relationships are highly significant (p<0.002) with R2 values varying between 0.30 and 0.76.

REVIEWER: ok

OK

**Comment 10**

Line 338: Figures 5 and 6 are difficult to interpret.

- A more detailed explanation of what these figures represent would improve clarity.

- What key insights should the reader take from these figures?

Authors: Yes, we agree that this was not well explained. In addition to what we wrote at lines 336 to 338, these figures also suggest that the results of the sensitivity analysis should be "reasonably well grounded in reality". We wrote this at line 343 in connection with figure 7, but this conclusion should also be based on figures 5 and 6 (now figure 6). In the revised version, we will modify the text at lines 380 to 389 to make this clearer. "A qualitative comparison with soil survey data for agricultural land in east- central Sweden (production area PO4) suggests that despite its simplicity the model estimates of steady-state SOC and bulk density in the soil profile lie mostly within the range of variation encountered in the region (Figure 6). Nevertheless, quantile-quantile plots show that the distributions of simulated and measured values of SOC and bulk density are different; especially at the tails, due to the much larger spread in the measurements compared with the calculations and especially the occurrence of a number of outliers with large values of organic carbon contents and small values of bulk density. This is not surprising because the calculations do not include the effects of all factors affecting SOC and bulk density. The large values of SOC (and small values of bulk density) almost certainly correspond to locations in the PO4 region with wet soils due to topography (i.e. flood plains, depressions). The model, as it is formulated here, does not include the effects of excess soil moisture on decomposition rates." We also removed the following sentence in the abstract, Line 23-24 ". The resulting model predictions compared well with aggregated soil survey data for the PO4 region" as this can be wrongly interpreted.

REVIEWER: Maybe the authors should propose the q-q plots as supplementary material.

OK, we have done so

**Comment 11**

Line 345: Minor inconsistency: (e.g., consistently use "Figure X" or "Fig. X rather than mixing "Fig. X" and "Figure X").

Authors: Yes, we fixed this in the revised version, we now use "Figure" at start of a sentence and "Fig." otherwise to match the SOIL journal guidelines.

REVIEWER: ok

OK

Response to comments from Reviewer 2

Authors: We would like to thank the Reviewer for the positive and perceptive comments, as well as for the questions, which will help us to improve the paper, specifically by giving more details, information and explanations about our data, methods and results.

This study describes a simple model of soil organic carbon (SOC) turnover that represents the effects of soil physical protection and microbial energy limitation. The paper first describes the model in a soil profile, then tests it using SOC data from a long-term study on agricultural fields with varied C inputs to the soils. Finally, the model's most influential parameters were identified in a sensitivity and uncertainty analysis. Overall, the paper is well written and presents a model of interest and relevance to soil carbon management. Below, please find specific comments intended to help improve the paper.

**Comment 1**

L83 – Define the abbreviation USSF. Authors: OK, we will do this in the revised version (it stands for Uppsala model of Soil Structure and Function), see line 93.

REVIEWER: ok
OK

**Comment 2**

L91-95 – I would suggest ending the introduction with a strong thesis statement of what the paper contributes to current knowledge of the subject.

Authors: Yes, we will add such a statement at the end of the introduction. See Lines 104-107 "The overall aim of this study is to demonstrate the utility of a simple soil C turnover model that can account for the nexus of soil management, soil structure and microbial activity that critically determines C mineralization and stabilization at the scale of a soil profile."

REVIEWER: the presentation of the purpose can still be improved, I suggest rephrasing the last paragraph entirely instead of only adding a final sentence. The research gap addressed by the authors should be mentioned at the end of the second last paragraph and connected with the general objectives of the study at the beginning of the last paragraph. The paper is composed of several phases, which are hard to follow. It is necessary to emphasize these phases in this last paragraph by adding "First,...", "second, ..." etc... You could also move the first lines of the M&M section in this last paragraph of introduction (see next comment).

OK, yes, we have followed this suggestion in re-writing the last part of the introduction in the revised version of the paper to clarify the research gap and the purpose of the study.

**Comment 3**

L97-101 – I appreciate this overview of the methods, very helpful to have this framework.
Authors: Thanks!

REVIEWER: ok. This overview (line 109-113) could be integrated in the last part of the intro for the sake of clarity.

There was already almost identical text in the last part of the Introduction at L92-100. It would lead to unnecessary repetition in the introduction if this text was incorporated. The introduction should now be much clearer after the latest revision (see the previous point).

We would like to keep this text here in the Materials and Methods section, as it gives some useful additional information on what the reader will find in the following sub-sections (as pointed out by the original referee nr. 1).

**Comment 4**

L103 – In section 2.1.1 that starts on this line, it is unclear if the model as described in this section is the work of the authors or if this is describing previously published work. If it has been previously published, I suggest including most of the equations in this section in a supplement rather than in the main document. In the main manuscript, I suggest describing the model in writing and including important equations for the modifications to the model that are new in the current study.

Authors: The model is based on a combination of the model described by Meurer et al. (2020) accounting for physical protection in relation to soil properties with the model described in Wutzler and Reichstein (2013) for microbial energy limitation. This combination of the two models was outlined in Coucheney et al. (2024), along with some minor improvements and modifications. However, the description of this SOM model was only included in the supplementary information in Coucheney et al. (2024) as the model itself was not tested at all. So although the main constituent components of the model have been described earlier, this is the first time that the complete model has been tested. As the model is quite new and previously untested, we prefer to keep these equations in the main paper, as it makes it easier for the reader. The paper is not too long and including the equations in the main text will ensure that the equations are readily available. We think this history of the model development is clearly explained at lines 104 to 115.

REVIEWER: I agree with the authors. In addition, presentation of equations helps to get the structure of the model which is critical for discussing parameters structural correlations.
OK

Additionally, I would encourage the authors to post their full model code online and cite it in the paper.
Authors: We built the model using the icon-based modelling software STELLA, which is a commercial product. The model file will be made available on request to the authors – this will be stated in the "data availability statement" at the end of the manuscript.

REVIEWER: ok
OK

**Comment 5**

107-109 – It would be helpful to specify the direction of the relationship between these effects (e.g., Do smaller pores get fewer root derived inputs?)

Authors: Not necessarily, no. It depends on the pore size distribution in the soil, which in turn depends on the soil texture (or more simply clay content in our approach). The pore size distribution determines the partitioning of root-derived inputs of OM between the two pore regions (see Eqs. 10-13). This means that a larger proportion of the root C inputs would enter the micropore region in a clay soil than in a sandy soil, because the porosity of a clay soil predominantly consists of smaller pores. Clay soils therefore have a higher potential for physical protection of soil C. The effect of the physical protection is quantified by the factor $F_p$ (Eqs. 1-4) that reduces the rate of SOC decomposition in micropores.

We will modify the text at Lines 119-124 to make this clearer: "In turn, the pore size distribution determines the partitioning of root-derived inputs of OM between the two pore regions. This means that compared with a sandy soil, a larger proportion of the root OM input will enter the micropore region in a clay soil, as it predominantly consists of smaller pores. The soil pore size distribution also regulates decomposition rates with slower decomposition rates of OM stored in microporous regions of the soil. Compared with sandy soils, clay soils therefore have a greater potential for physical protection of soil C.".

REVIEWER: ok
OK

Do micropores have lower decomposition rates?).
Authors: Yes, they do. We will clarify this in the revised text at lines 107-109 (see above text)

REVIEWER: ok
OK

**Comment 6**

L125 – Is the "(-)" after $f_{r,mic}$ supposed to indicate that it is unitless?
Authors: Yes

REVIEWER: ok
OK

**Comment 7**

L218 – How does this straw addition rate compare to the maize biomass per hectare?
Authors: Note that maize was only grown since 2000. Between 2000-2019, the crop fertilized and crop fertilized+straw treatments had on average 6.07 and 7.09 t ha-1 maize yield, respectively (see Kätterer & Bolinder, 2024). However, above-ground maize and crop residues were removed from the field and it is very speculative how much C is added to the soil system via root inputs (rhizodeposition and exudates) and such C inputs are challenging to quantify and are therefore not included in the present proposed model. For the modelling, we know how much straw was added and this is included in the paper.

REVIEWER: ok
OK

**Comment 8**

L219 - 220 – Include the scientific names/varieties of the crops.

Authors: We will include Latin names in the revised version (lines 242-244): "Maize (Zea mays) has been grown on the cropped plots since 2000. Before 2000, the crop rotation included barley (Hordeum vulgare), oats (Avena sativa), beets (Beta vulgaris) (prior to 1967) and rape (Brassica napus)."

In addition, we propose to include a reference to a newly published data paper which gives a complete description of the experiment, as well as links to all the data (Data in Brief paper. Pold et al. 2025 - Soil and vegetation property data from the Ultuna R3-RAM56 long-term soil amendment experiment, 1956-2023), at line 233.

REVIEWER: ok

OK

**Comment 9**

L220 - What is the purpose of hand digging the plots after harvest?

Authors: It's to simulate ploughing: the plot size of 4 m2 is too small to manage in the same way as a farmer's field. We will add this information at Line 244: "to simulate ploughing as the plots are too small (4 m2) to be managed in the same way as a farmer's field."

REVIEWER: ok
OK

**Comment 10**

L 231 – What measurements were used for the calibration? Only OM or additional measurements?

Authors: Only SOC. We will state this explicitly in the revised version at line 254.
REVIEWER: ok

OK

Calibrating to additional variables could help reduce equifinality.
Authors: Perhaps, depending on the type of data and its quantity and quality. But we were able to strongly reduce the prior uncertainty ranges for the parameter values anyway, despite parameter correlation.  See also our answer to comment number 7 from Reviewer1

REVIEWER: ok.
OK

**Comment 11**

L238 – Specify the field bulk density values used for this validation.

Authors: OK, yes, we will do so (the numbers are 1.43, 1.28 and 1.21 g/cm3 in the treatments "Fallow", "N-fertilized" and "N fertilized + straw" respectively). This was added at lines 276-277.

REVIEWER: ok
OK

**Comment 12**

Table 1 – Are there field measurements available for any of these parameters? If so, how similar are they to these fixed values?

Authors: Yes, we didn't write it explicitly but these values in table 1 were based (with one exception) on data obtained at the study site. We will add a column to this table with the heading "Source" where we will cite the relevant studies.

See also answer to comment 4 – Reviewer1

REVIEWER: ok

OK

**Comment 13**

Figure 1 needs a legend to identify what the different colors/patterns of shading indicates.

Authors: We will add this information to the caption (at lines 227-229) "Figure 2. Map of Sweden (in white) showing the location of the Ultuna Long-term Soil Organic Matter Experiment (Uppsala, Sweden) and the extent of the production area PO4 (shaded area in grey). Drawn by Anna Lindahl, SLU, from Esri, TomTom, Garmin, FAO, NOAA, and USGS"

REVIEWER: ok. What does PO4 stand for? You could indicate it from the beginning.

PO stands for "produktionsområde" i.e. production area in Swedish. So it's production area number 4. In the revision, we have deleted PO to avoid any confusion

**Comment 14**

Table 1 – Are there data references for these parameter values? If not, how did you come to these values?

Authors: Yes, we will give these references. We will add a column to this table with the heading "Source"

REVIEWER: The question was related to Table 1, not to Table 2. Could you please explain also for Table 1 how you come to these values?

This is a misunderstanding. We did make the changes that the earlier referee requested. The confusion has arisen because as a consequence of meeting the comments of referees 1 and 2, the numbering of tables 1 and 2 changed in the revised version of the paper (Table 1 became Table 2 and vice versa). It is understandable that referee 3 did not realize this.

Tables – I recommend including captions for tables with relevant details.

Authors: We will revise the table captions giving more informative detail. For example:

Lines 278-279: "Table 1. Six model parameters selected Initial parameter uncertainty ranges for the model calibration to the Ultuna Long-Term Soil Organic Matter Experiment and their initial parameter uncertainty ranges"

Lines 281-282: "Table 2. Nine model parameters fixed at constant values during the calibration based on field measurements at Ultuna or literature data"

Page 15: "Table 3. Parameter input distributions in the sensitivity analysis. In the case of uniform distributions, minimal and maximal values are shown (Min.; Max.) while in the case of normal distribution the mean and standard deviation are shown (Mean; St. dev.)."

REVIEWER: ok

OK

**Comment 15**

L264- What determined if the data support was sufficient? Authors: This is partially a subjective decision. However, we can revise the sentence to be a bit more specific:

Lines 303-305: "We assumed normal distributions when the data support was considered sufficient to support such a distribution, while uniform distributions were used otherwise (Table 3) "

REVIEWER: ok

OK

**Comment 16**

Table 3 – The source "SCB Statistics Sweden" needs to be more specific. Same comment for source listed as "site data" - where are site data accessible? Year, dataset name, authors etc. Authors: Yes, we have added these details.

REVIEWER: ok

OK

**Comment 17**

L304 - 306 – Is this distinction of straw going into only mesopores and root OM going partially into micropores supported by empirical data?

Authors: No, it's more a model hypothesis, although based on process understanding. We can't see how digging or ploughing down above-ground crop residues could possibly incorporate these residues into the microporous regions of the soil where pore diameters are less than 5 microns (in this study). In contrast, roots can grow through microporous soil regions supplying both POM (on root death) and root exudates.

This hypothesis could be tested experimentally in future work with for example, the help of X-ray tomography on samples taken soon after harvest and ploughing-down of above-ground crop residues). This is however way beyond the scope of the present study.

REVIEWER: Could you please explain this in the text?

OK, yes, we have added an explanation, in connection with the section describing the model, which is where it best belongs. We also added another new sentence to the model description which should help the reader better understand its mechanistic basis.

**Comment 18**

L308 – What is meant by "export of residues?" Does that mean the removal of residues by land managers?

Authors: Yes. In this treatment, most of the above-ground crop residues are removed. This was explained at line 217. We will replace this phrase by , at line 350: "due to the near total removal of above-ground crop residues."

REVIEWER: ok

OK

**Comment 19**

Figure 3 – This panel figure needs letters for each panel and a description of each panel and the definition of the X axes in the figure caption.

Authors: We feel that letters for the panels are not needed but we will give further details in the figure caption instead, page 22 (now figure 4):

"Figure 4. Mean model efficiencies for each parameter set (only simulations with model efficiencies larger than zero are shown) plotted against the values for the six parameters in the GLUE analysis (refer to table 1 for parameter definitions and descriptions; OM = organic matter, AG = above-ground)"

REVIEWER: ok. To make the figure understandable on its own, could you briefly recall how it was generated? L355: Explain for non-GLUE users and non-statisticians why we can graphically conclude from new Fig 4 that epsilon is constrained, while the other parameters are not.

This was explained in the paper, at the place where figure 4 was first introduced (L388):
*"Figure 4 shows that only one of the parameters included in the calibration procedure (the OM retention coefficient, $\varepsilon$) was well constrained by the data, with acceptable values lying within a narrow range (ca. 0.30 to 0.35). In contrast, for the other five parameters, simulations with large model efficiencies could be found across almost the entire prior uncertainty ranges (Figure 4)"*

Figure 4 – This panel figure also needs to have the individual panels labeled/described and the axes defined in the caption.

Authors: We feel that letters for the panels are not needed because we don't refer to individual subfigures in the text, but we will give further details in the figure caption instead, page 24 (now figure 5): "Figure 5. Inter-relationships among four of the six model parameters included in the calibration procedure. Relationships are shown for the 30 best parameter sets identified in the GLUE analysis (refer to table 1 for parameter definitions and descriptions)."

REVIEWER: Thank you, but please provide the meaning of the parameters instead of asking to refer to Table 1

OK, we have done so

**Comment 20**

L314 – In the introduction, large mechanistic models are criticized for having uncertainty and equifinality. That was provided as justification for a simpler parsimonious model. How does that criticism relate to your finding that the simple/parsimonious model presented here has the same issue of equifinality and parameter uncertainty as the more complex models? How does this affect the usefulness of the model or its applicability compared to the larger models? Or, should this model's parameters be further simplified?

Authors: Please see our answer to comment 2 of Reviewer1 (also copied here) Yes, this is a good point. It's true that even the simplest models of organic matter turnover in soil can show equifinality, depending on the type, quantity and quality of the data used to constrain them. This has been demonstrated for models that are even simpler than the one we developed and tested in our study (see e.g. Juston et al., 2010; Luo et al., 2017). We were therefore expecting to encounter the issue, as we wrote at line 233. This is also why we wrote "may" on line 48.

However, in contrast to more complex models, we can clearly see why and where the equifinality arises in our relatively simple model: it depends on the model structure, with correlations among only a few parameters, which makes the problem of parameter uncertainty more manageable. This was the case for the model application to the data at the Ultuna long-term soil organic matter experiment presented in the paper: here, simultaneous calibration to data was sufficient to effectively constrain the model parameters in three treatments with strongly contrasting inputs of OM with respect to both type and amount. This is what we concluded at lines 317-318: "These strong correlations of ko, Aa and ktill with epsilon mean that, in practice, all four parameters are well constrained by the calibration".

REVIEWER: ok

OK

**Comment 21**

L317 – What is the evidence of strong correlation? This statement needs statistical support.
Authors: Yes, we now added R2 and p-values in a revised version of the new figure 5. All four relationships are highly significant (p<0.002) with R2 values varying between 0.30 and 0.76.

REVIEWER: ok

OK

**Comment 22**

L341 Temperate, not temperature.

Authors: Thanks, we fixed this

REVIEWER: ok

OK

**Comment 23**

Figure 5 – Can you provide a quantitative comparison of the means to support the conclusions?

Authors: This is a good point. We didn't attempt any statistics, simply writing that the model gives reasonably realistic predictions". In fact, statistical tests for differences between the two distributions (not just the means) show that they are significantly different, which is almost entirely due to the much larger spread in the measurements compared to the calculations, especially the occurrence of a number of outliers with large values of organic carbon contents and correspondingly small values of bulk density. This is not really surprising because the calculations do not include the effects of all factors affecting SOC and bulk density. Our guess is that the large values of organic carbon content (and small values of bulk density) correspond to locations with wet soils due to topography (i.e. flood plains, depressions). The model, as it is formulated here, does not include the effects of excess soil moisture on decomposition rates.

We have added some new text in the revised version at lines 380-389 to explain the above. "A qualitative comparison with soil survey data for agricultural land in east-central Sweden (production area PO4) suggests that despite its simplicity the model estimates of steady-state SOC and bulk density in the soil profile lie mostly within the range of variation encountered in the region (Figures 6 and 7). Nevertheless, quantile-quantile plots show that the distributions of simulated and measured values of SOC and bulk density are different; especially at the tails, due to the much larger spread in the measurements compared with the calculations and especially the occurrence of a number of outliers with large values of organic carbon contents and small values of bulk density. This is not surprising because the calculations do not include the effects of all factors affecting SOC and bulk density. The large values of SOC (and small values of bulk density) almost certainly correspond to locations in the PO4 region with wet soils due to topography (i.e. flood plains, depressions). The model, as it is formulated here, does not include the effects of excess soil moisture on decomposition rates."

We will avoid using the word "predictions", because these are the aggregated outputs of a sensitivity analysis and not model predictions that can be compared with measurements at specific locations.

REVIEWER: Please add q-q plots as supplementary

OK, we have done so

**Comment 24**

Figure 6 – Why are only two depths shown in this figure? (there are 3 depths in the previous figure)

Authors: Because we only have bulk density data at two depths, i.e. between 0-20 cm depth and 40-60 cm depth, whereas SOC was measured between 0-20, 20-40 and 40-60 cm depth (see line 281). This appears better now that both figures have been merged under the name Figure 6 (see page 25)

REVIEWER: ok
OK

L357 – What are the cutoffs for NRC values to determine if they are strongly, moderately, or minimally sensitive in this analysis? I think that information should be included in the methods. In the discussion, it would be helpful to more quantitatively compare/describe the model sensitivity to these various parameters. Authors: We prefer to discuss these results in relative terms and therefore we choose to present all values ordered decreasingly in Table 5 without any use of defining cut off values that may be judged to be arbitrary (the readers can also see all of the figures and apply their own criteria if wanted)

REVIEWER: the current version of the paper is understandable; however, it is necessary to indicate the ranking method in the table legend.
We realize now that we used different symbols in the table and the equation describing the method. We used NRC in the table caption, but β in equation 26 describing the method. We apologize for this and have fixed this in the revised version.

**Comment 25**

L359 – The fraction of aboveground residues incorporated is roughly as important as the clay content yet it is not mentioned here.

Authors: Yes, we agree that it should also be mentioned. We will include a mention of finc at line 418, alongside the other two parameters that determine the input of above-ground residues.

REVIEWER: ok

OK

I'm also unclear why the clay content is mentioned before the other more sensitive parameters in the table.

Authors: Yes, we see your point. We mentioned the clay content here because it belongs together with Fp in that taken together they determine the extent of physical protection. We did try to explain this at lines 414-416, but we will make this connection much clearer in the revised version:

"The soil clay content, which together with Fp, determines the extent to which physical protection is expressed in soils of contrasting texture, is also a relatively sensitive model parameter (Table 5)"

REVIEWER: ok

OK

**Comment 26**

L367-370 – I would be interested to see this idea expanded upon – how does this USSF model result relate back to your model result of an 8% increase in SOM?

Authors: It's really mostly just a consequence of the fact that it takes a lot longer than 30 years to reach steady-state after a change in OM inputs. We added the word "shorter" to highlight this at line 424.

REVIEWER: ok

OK

And what are the larger implications of these increases for climate change mitigation as you mention? For example, how does a 1.4% increase over the course of 30 years compare to targeted goals for mitigation?

Authors: This is discussed in Coucheney et al. (2024) and we refer to this study in the text. We don't think this is the right place to discuss the results from a previous paper.

REVIEWER: ok

OK

**Comment 27**

L379 – Here, the authors seem to consider a 4-5% reduction in SOM to be minor. But on lines 374 - 377 they seem to indicate that a 3-5% increase is significant. Some benchmarks for the relevance of these changes would be helpful for interpretation of the results.
Authors: Thanks for pointing this out. It is only an apparent inconsistency, because the 3-5% increase mentioned at lines 374-377 was after 30 years, whereas at line 379 we are referring to steady-state values. We will make this clearer by slightly modifying the sentence at L433-434: "Table 5 suggests that tillage is one of the least sensitive factors affecting SOM stocks at steady- state: "

REVIEWER: ok

OK

**Comment 28**

L381 – Haddaway et al. found a difference by tillage intensity in the topsoil, as opposed to what is stated here. Intermediate intensity tillage resulted in greater SOC stocks than the high intensity tillage in that metanalysis.

Authors: This seems to be a misunderstanding. We did write that Haddaway et al. found greater stocks of C in the topsoil under no-till. However, we also wrote that they did not find any difference in total C stocks in the profile between no-till and intensive or intermediate tillage systems, which is also true. We will slightly revise the sentence at lines 437-440: "…. Haddaway et al. (2017) and Meurer et al. (2018) only found larger SOC stocks under no-till compared with conventional tillage in the topsoil, but no significant differences in total SOC stocks in these two tillage systems in soil profiles to 60 cm depth."

REVIEWER: ok

OK

**Comment 29**

L385 – It would be helpful to include empirical data supporting these model data on the figure for comparison with the model data. Or include the empirical data in a table caption.

Authors: We would prefer not to modify the actual figure to show data from other studies. This is partly because a direct comparison of predictions for a region in east-central Sweden with measurements from a highly diverse global dataset like Chen et al. (2020) could give readers a misleading impression. In addition, the raw data on MRT from the study by Poeplau et al. (2021) are not available to us. The value quoted in the paper at line 339, "ca. 20 years" was taken from a table in the paper (table 1). In the revised version of the paper, we will also add the values reported by Chen et al. (2020) for cropland to this text, as this data is available. In doing so, we will also slightly modify the text at lines 395 to 399: "…. and they also lie at the high end of the range in the global analysis reported by Chen et al. (2020) for croplands (mean = 9.5 years, standard deviation = 6 years, n = 217). Taken together with Fig. 6, this gives us confidence that the results of the sensitivity analysis presented in the following should be reasonably well grounded in reality." We feel this is better than adding these numbers to the figure caption.

REVIEWER: How was MRT calculated?
This was stated at L295: *"For each soil horizon, we also calculated the steady-state bulk density and SOM contents as well as the mean residence time of SOM as the steady-state SOM stock divided by the input/output flux."*

You could discuss the prediction with MRT over the soil profile published by Balesdent et al in 2018 in Nature.

Yes. Our simulated MRT values are quite similar to the median age estimates in their data. We have referred to the results in this paper in the revised version.

**Comment 30**

Table 5 – It looks like this table was color coded according to the colors of the groups in table 3. That information and the meaning of the colors should be included in the caption.
Authors: Yes, that's right. However, we have been advised by the editor that we must remove this colour coding in the revised version of the paper.

REVIEWER: ok

OK

**References cited (in answers to both Reviewer 1, 2 and 3)**

Andrén, O., Kätterer, T. 1997. ICBM: the introductory carbon balance model for exploration of soil carbon balances. Ecological Applications, 7, 1226-1236.

Chen, S., Zou, J., Hu, Z., Lu, Y. 2020. Temporal and spatial variations in the mean residence time of soil organic carbon and their relationship with climatic, soil and vegetation drivers. Global and Planetary Change, 195, 103359.

Coucheney, E., Kätterer, T., Meurer, K.H.E., Jarvis, N. 2024. Improving the sustainability of arable cropping systems by modifying root traits: a modelling study for winter wheat.European Journal of Soil Science, 75, e13524. Dechow, R., Franko, U., Kätterer, T., Kolbe, H. 2019. Evaluation of the RothC model as a prognostic tool for the prediction of SOC trends in response to management practices on arable land. Geoderma, 337, 463-478.

Kätterer, T., Bolinder, M. 2024. Response of maize yield to changes in soil organic matter in a Swedish long-term experiment. European Journal of Soil Science, 75, e13482. Juston, J., Andrén, O., Kätterer, T., Jansson, P-E. 2010. Uncertainty analyses for calibrating a soil carbon balance model to agricultural field trial data in Sweden and Kenya. Ecological Modelling, 221, 1880-1888.

Luo, Z., Wang, E., Sun, O. 2017. Uncertain future soil carbon dynamics under global change predicted by models constrained by total carbon measurements. Ecological Applications, 27, 1001-1009.

Meurer, K., Chenu, C., Coucheney, E., Herrmann, A., Keller, T., Kätterer, T., Nimblad Svensson, D., Jarvis, N. 2020. Modelling dynamic interactions between soil structure and the storage and turnover of soil organic matter. Biogeosciences, 17, 5025-5042 Poeplau, C., Don, A., Schneider, F. 2021. Roots are key to increasing the mean residence time of organic carbon entering temperate agricultural soils. Global Change Biology, 27, 4921– 4934.

Pold, G., MacDonald, E., Braun S., Herrmann, A. M., 2025. Soil and vegetation property data from the Ultuna R3-RAM56 long-term soil amendment experiment, 1956-2023

Wutzler, T., Reichstein, M. 2013. Priming and substrate quality interactions in soil organic matter models. Biogeosciences, 10, 2089-2103.